

# Marginal deformations from type IIA supergravity

**Nikolay Bobev[1], Pieter Bomans[2,3]\*, Fridrik Freyr Gautason[4] and Vincent S. Min[1]**

**1** Instituut voor Theoretische Fysica, K.U. Leuven,
Celestijnenlaan 200D, BE-3001 Leuven, Belgium
**2** Dipartimento di Fisica e Astronomia, Università di Padova,
via Marzolo 8, 35131 Padova, Italy
**3** INFN, Sezione di Padova, via Marzolo 8, 35131 Padova, Italy
**4** Science Institute, University of Iceland, Dunhaga 3, 107 Reykjavík, Iceland

\* pieter.bomans@pd.infn.it

## Abstract

We study marginal deformations of a class of three-dimensional $\mathcal{N} = 2$ SCFTs that admit a holographic dual description in massive type IIA supergravity. We compute the dimension of the conformal manifold in these SCFTs and identify a special submanifold with enhanced flavor symmetry. We show how to apply the TsT transformation of Lunin-Maldacena to construct explicit supersymmetric AdS$_4$ solutions of massive type IIA supergravity with non-trivial internal fluxes that are the holographic dual of these marginal deformations. Finally, we also briefly comment on a class of RG flows between some of these SCFTs that also admit a holographic realization.


## 1  Introduction

Generic CFTs in more than two dimensions do not possess exactly marginal operators and are therefore isolated. The existence of exactly marginal operators seems to require some "fine-tuning" and is expected to be associated with an underlying symmetry, see [1–3] for a recent discussion. Indeed, all examples of CFTs with a finite number of degrees of freedom that have exactly marginal operators arise in 3d and 4d theories invariant under at least four supercharges. This special status of exactly marginal operators calls for a more systematic understanding of their properties. The geometry of the space of exactly marginal couplings, known as the conformal manifold, is particularly interesting and in general challenging to explore quantitatively. In 4d SCFTs with at least $\mathcal{N} = 1$ supersymmetry one can engineer many examples of theories with exactly marginal operators. In many of these examples, the existence of the marginal operators is related to the fact that the Yang-Mills coupling in a non-Abelian gauge theory is classically marginal and quantum corrections to its anomalous dimension may cancel due to supersymmetry [4]. This leads to the favorable situation in which the conformal manifold has a weakly coupled region which can be accessed by employing perturbation theory, see for example [5] for a recent discussion and [6] for some explicit examples. In general, however, conformal manifolds can be compact, i.e. there may not be any weakly coupled regime accessible by perturbation theory. This situation is bound to arise in 3d SCFTs with $\mathcal{N} = 2$ supersymmetry since the Yang-Mills coupling is irrelevant while Chern-Simons couplings for the gauge fields are quantized. Indeed, there are very few explicit results for the physics of conformal manifolds in 3d $\mathcal{N} = 2$ Chern-Simons-matter theory, see [7–9] for some examples. Our goal here is to study conformal manifolds in a class of examples arising from D2-branes in massive IIA string theory by utilizing their dual supergravity description.

Using supergravity and holography to understand the conformal manifold of SCFTs arising from string theory is a subject with a long history, see for instance [10–12]. However, even for the well-known 4d $\mathcal{N} = 4$ SYM theory there are many open questions. This theory has a three-dimensional complex conformal manifold along which $\mathcal{N} = 1$ supersymmetry is preserved. Two of the directions on this manifold are parameterized by superpotential deformations and one direction is given by the complexified gauge coupling. On a generic point of this conformal manifold, the theory has only $U(1)_R$ R-symmetry and no continuous flavor symmetry. The family of AdS$_5$ string theory backgrounds corresponding to this three-dimensional conformal manifold is not known and is expected to be difficult to construct explicitly since the SO(6) isometry of the internal $S^5$ will be broken to U(1). In [11] Lunin and Maldacena made headway on this problem by focusing on the so-called $\beta$-deformation of the superpotential of $\mathcal{N} = 4$ SYM which preserves U(1) × U(1) flavor symmetry in addition to the $U(1)_R$. Using a series of duality transformations of string theory, known as the TsT transformation, they deformed the well-known AdS$_5 \times S^5$ solution into a one-parameter family of explicit AdS$_5$ supergravity backgrounds with internal fluxes that are dual to a submanifold of the SCFT con-

formal manifold. The main focus of this note is to generalize this procedure to $AdS_4$ solutions in massive IIA supergravity dual to 3d $\mathcal{N} = 2$ Chern-Simons-matter SCFTs.

The class of SCFTs we study is given by 3d $\mathcal{N} = 2$ Chern-Simons theories with multiple $SU(N)$ (or $U(N)$) gauge groups coupled to bifundamental and adjoint chiral multiplets. We assume that the sum of the Chern-Simons couplings for all gauge groups does not vanish. These theories, therefore, break parity and in the large $N$ limit are candidate holographic duals to supersymmetric $AdS_4$ solutions in massive IIA supergravity [13,14]. Indeed, explicit $AdS_4$ backgrounds of this class were constructed in [15] and [16]. Convincing evidence for the validity of the holographic duality was presented in [15, 16] by using supersymmetric localization results for the $S^3$ free energy which, in the large $N$ limit, precisely agree with the supergravity on-shell action. A notable feature of these models is that the $S^3$ free energy scales as $n^{1/3} N^{5/3}$, where $n$ is the sum of the Chern-Simons levels. Moreover, it was argued in [16] that each of these 3d $\mathcal{N} = 2$ Chern-Simons quiver gauge theories can be thought of as arising from a "parent" 4d $\mathcal{N} = 1$ gauge theory with the same matter content and superpotential. If the 4d $\mathcal{N} = 1$ SCFT admits an $AdS_5 \times Y^5$ supergravity dual, then one can use the details of the Sasaki-Einstein manifold $Y^5$ to construct the explicit massive IIA $AdS_4$ supergravity solution dual to the 3d $\mathcal{N} = 2$ SCFT. Guided by these results we study several examples of these models given by "necklace quivers" which have supergravity dual solutions controlled by $Y^5 = S^5$, $Y^5 = T^{1,1}$ and orbifolds thereof. Using the explicit superpotential of these theories and their global symmetries we apply the results in [17–19] to compute the dimension of their conformal manifold. We then identify a subspace of this manifold along which the exactly marginal couplings preserve $U(1) \times U(1)$ flavor symmetry. We then turn on to the supergravity side where we use the supergravity solutions in [15,16] as a starting point on which to apply the Lunin-Maldacena TsT transformation. The result is a one-parameter family of supergravity solutions which should be dual to the exactly marginal deformation in the dual SCFT. We provide some evidence for this claim by showing that the flux quantization of the supergravity solution, as well as the holographic calculation of the $S^3$ free energy, are not modified by the TsT transformation despite the very non-trivial changes in the metric and internal fluxes.

We start in the next section by introducing the type of 3d SCFTs under consideration and discuss in detail several examples with a particular focus on their exactly marginal deformations. In Section 3 we start by introducing a set of seed $AdS_4$ supergravity solutions dual to the undeformed SCFTs and then describe the result of the TsT transformation to these solutions and its relation to the SCFT marginal deformations. In Section 4, we conclude with a discussion of several interesting open problems and possible generalizations. The three appendices contain our supergravity conventions as well as some of the technical details related to the TsT transformation.

## 2 Field theory

We are interested in 3d $\mathcal{N} = 2$ Chern-Simons-matter of quiver theories with adjoint and bifundamental chiral multiplets. We will focus on theories that have non-vanishing sum of the Chern-Simons levels for the gauge groups and thus they break parity. It turns out that this class of theories can be engineered in string theory by considering D2-branes in the presence of non-trivial Romans mass, i.e. D8-brane flux. Moreover as pointed out in [15, 16] these SCFTs bare close resemblance to a large class of 4d $\mathcal{N} = 1$ quiver gauge theories that arise on the worldvolume of a stack of D3-branes probing a Calabi-Yau singularity $X$. This connection can be understood qualitatively by considering the brane setup in more detail (see [20,21]). Indeed, let us start with a system of $N$ D3-branes on the background $\mathbf{R}^{1,3} \times X$, where $X$ is a local Calabi-Yau three-fold singularity. The low energy effective theory living on this system

of branes probing the Calabi-Yau singularity is expected to be given by a four-dimensional $\mathcal{N} = 1$ quiver gauge theory. For example when $X$ is a toric Calabi-Yau singularity, admitting a (crepant) resolution $\widetilde{X}$, the low energy theory is given by a $U(N)^{\chi(\widetilde{X})}$ quiver gauge theory, where $\chi(\widetilde{X})$ is the Euler characteristic of $\widetilde{X}$, with a particular superpotential. Performing a T-duality transforms the D3-brane system into a system of D2-branes probing $\mathbf{R} \times X$, whose worldvolume theory is given by a 3d $\mathcal{N} = 2$ quiver gauge theory. This 3d theory has exactly the same quiver representation and superpotential as the four-dimensional one. In the following, we will refer to the 4d theory as the "parent" and the 3d one as the "daughter" theory. The freedom to add Chern-Simons couplings for the gauge groups in the quiver gauge theory is left unaccounted for in this duality transformation. In the class of models studied in [15, 16] all Chern-Simons levels are equal, $k_a = k$. As explained in [14], in massive type IIA string theory, this corresponds to adding a Romans mass $F_0 = \frac{n}{2\pi\ell_s}$, where $\ell_s$ is the string length and $n = \chi(\widetilde{X})k$ is the sum of Chern-Simons levels. Here we will be interested in the properties of these SCFTs in the large $N$ limit with a particular focus on their exactly marginal deformations. We will describe several illustrative examples of such quiver SCFTs below in preparation for Section 3 where we discuss their supergravity dual.

## 2.1 Examples

**One-node quiver**

Using this brane intuition, various examples of such three-dimensional SCFTs were studied in [15, 16]. Here we will highlight some particular examples and discuss their conformal manifold and various RG flows relating them. As a first example, consider the three-dimensional daughter theory associated to four-dimensional $\mathcal{N} = 4$ SYM [15]. This is a Chern-Simons-matter theory with gauge group $SU(N)$ at level $k$ coupled to three chiral superfields, $X_i$, in the adjoint representation of the gauge group, see Figure 1. The superpotential of this theory is given by

$$\mathcal{W} = \text{Tr}\left(X_1[X_2, X_3]\right). \tag{1}$$

The theory has $SU(3)_F \times U(1)_R$ global symmetry, where $SU(3)_F$ is a flavor symmetry which rotates the three chiral superfields $X_i$ and $U(1)_R$ is the superconformal R-symmetry. The three chiral superfields have R-charge $q_R = 2/3$ and dimension $\Delta_{X_i} = 2/3$.[1]

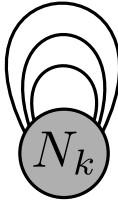

Figure 1: The quiver of 4d $\mathcal{N} = 4$ SYM (left) interpreted as a 3d $\mathcal{N} = 2$ quiver with Chern-Simons level $k$.

In order to study the conformal manifold, we start by listing the candidate marginal operators of the theory. These are given by cubic gauge invariant operators of the form $h^{ijk}\text{Tr}(X_iX_jX_k)$ which have R-charge 2 and thus conformal dimension $\Delta = 2$. Each of these is a chiral primary operator which is the lowest component of a multiplet containing an operator of dimension $\Delta = 3$. After taking into account the chiral ring relations, there are 10 independent cubic chiral operators which can be decomposed as the $\mathbf{8} \oplus \mathbf{1} \oplus \mathbf{1}$ representation of $SU(3)_F$.[2] As

---

[1]As usual we specify the R-charge of the complex scalar in the chiral superfield.

[2]This counting agrees with the recent calculation of the perturbative KK spectrum around the $AdS_4$ solution of type IIA supergravity dual to this SCFT [22].

emphasized in [17–19] not all of these operators are exactly marginal. To compute the number of exactly marginal operators, i.e. the dimension of the conformal manifold, we employ the formalism of [17–19]. It was shown there that one should subtract the dimension of the continuous flavor group from the naive number of marginal operators to arrive at the final number of exactly marginal operators. Applying this to the example at hand we conclude that the complex dimension of the conformal manifold, $\mathcal{M}_c$, of this three-dimensional $\mathcal{N} = 2$ SCFT is[3]

$$\dim_{\mathbf{C}}(\mathcal{M}_c) = 10 - \dim(SU(3)_F) = 2 \,. \tag{2}$$

Our main interest in this note is a particular one-dimensional subspace of this two-dimensional conformal manifold. Namely, the unique exactly marginal deformation preserving a $U(1)^2$ subgroup of $SU(3)_F$. This deformation can be parameterized by a complex number $\beta$ which deforms the superpotential in (1) to

$$\mathcal{W} = \mathrm{Tr}\left(e^{i\pi\beta} X_1 X_2 X_3 - e^{-i\pi\beta} X_1 X_3 X_2\right) \,. \tag{3}$$

The other exactly marginal deformation, parameterized by the complex coupling $\lambda$, which breaks the $U(1)^2$ flavor symmetry can be thought of as arising from the following superpotential deformation

$$\delta\mathcal{W} = \lambda \mathrm{Tr}\left(X_1^3 + X_2^3 + X_3^3\right) \,. \tag{4}$$

Note that these exactly marginal deformation are very similar to the ones of four-dimensional $\mathcal{N} = 4$ SYM [4,10,11]. A notable difference is that in four dimensions there is one additional exactly marginal deformation given by the complexified gauge coupling which is clearly not a marginal coupling in a three-dimensional gauge theory.

In general, it is extremely hard to compute physical observables in this strongly coupled SCFT. However, supersymmetric localization allows the exact evaluation of some BPS observables. For example, the free energy on $S^3$ of the SCFT at hand can be computed using the results in [23,24]. In the large $N$ limit, the real part of the free energy is scheme independent and is given by [15]

$$\mathrm{Re}\,\mathcal{F}_{\mathcal{N}=4} = \frac{2^{1/3} 3^{1/6}}{5} \pi \, k^{1/3} N^{5/3} \,. \tag{5}$$

Superpotential marginal deformations are $Q$-exact with respect to the localization supercharge and therefore the free energy remains constant on the entire conformal manifold and takes the value in (5).

We can obtain another SCFT from this setup by deforming the theory discussed above by a relevant deformation. This is analogous to the four-dimensional Leigh-Strassler (LS) RG flow discussed in [4] and the relevant deformation is given by adding $\mathrm{Tr}\left(X_1^2\right)$ to the superpotential in (1). The quiver for this model is obtained from Figure 1 by erasing one of the lines. In the IR one obtains a strongly coupled theory with a quartic superpotential for the two chiral superfields $X_{2,3}$. Naively this theory has an $SU(2)_F$ flavor symmetry under which $X_{2,3}$ transform as a doublet and a $U(1)_R$ symmetry under which $X_{2,3}$ have charge $1/2$. A more careful analysis using both QFT and holography methods suggests that this SCFT enjoys an enhancement of supersymmetry to $\mathcal{N} = 3$ [8,15,25,26]. Therefore the $U(1)_R$ symmetry is actually enhanced to $SO(3)_R$. The marginal operators in this SCFT are given by supersymmetric descendants of holomorphic quartic operators built out of $X_{2,3}$. There are in total 5 such operators that transform in the $\mathbf{3} \oplus \mathbf{1} \oplus \mathbf{1}$ of the $SU(2)_F$.[4] The number of exactly marginal operators, i.e. the

---

[3]Alternatively, one could use the method of [4] to compute the dimension of the conformal manifold. However, since the three-dimensional theory at hand is not weakly coupled the computation of the $\beta$-functions is non-trivial.

[4]Again, this counting is in harmony with the KK spectroscopy for the AdS$_4$ solution of type IIA supergravity dual to this SCFT [22].

complex dimension of the conformal manifold, can again be computed as we did above

$$\dim_{\mathbf{C}}(\mathcal{M}_c) = 5 - \dim(\mathrm{SU}(2)_F) = 2 \,. \tag{6}$$

This result was also obtained in [8] where this model was studied in some detail. We therefore conclude that the UV and IR SCFTs have conformal manifolds of equal dimensions. We note that the $\mathcal{N} = 3$ supersymmetry is broken to $\mathcal{N} = 2$ on a generic point of this conformal manifold. The continuous flavor symmetry is generically broken, however, there is a one-dimensional complex submanifold of the conformal manifold along which $\mathrm{U}(1)_F \subset \mathrm{SU}(2)_F$ is preserved. The superpotential on this one-dimensional submanifold of $\mathcal{M}_c$ can be written as

$$\mathcal{W} = \mathrm{Tr}\left(e^{i\pi\beta}X_2 X_3 X_2 X_3 - e^{-i\pi\beta}X_2 X_3 X_3 X_2\right) \,. \tag{7}$$

The $S^3$ free energy of this IR theory, which we denote by LS, is given by [27]

$$\mathrm{Re}\,\mathcal{F}_{\mathrm{LS}} = \frac{9 \times 3^{1/6}}{40}\pi\,k^{1/3}N^{5/3} \,, \tag{8}$$

and is again the same on the whole conformal manifold. We note that the $S^3$ free energy of the IR and UV SCFTs are related as

$$\frac{\mathrm{Re}\,\mathcal{F}_{\mathrm{LS}}}{\mathrm{Re}\,\mathcal{F}_{\mathcal{N}=4}} = \left(\frac{27}{32}\right)^{2/3} \,. \tag{9}$$

We comment further on this in Section 2.2 below.

**Two-node quiver**

The second example we would like to study is a quiver with two nodes which is identical to the quiver corresponding to the $\mathbf{Z}_2$ orbifold of the $\mathcal{N} = 4$ SYM theory. This model preserves half of the maximal supersymmetry in four dimensions but from the 3d perspective studied here it has $\mathcal{N} = 2$ supersymmetry. This three-dimensional quiver gauge theory contains two $\mathrm{U}(N)$ gauge groups with equal Chern-Simons level $k$, see Figure 2. The field content of this SCFT consists of two adjoint vector multiplets, two adjoint chiral multiplets, $\Phi_{1,2}$, and four bi-fundamental chiral superfields. Two of the chiral superfields, $A_{1,2}$, transform in the $(\mathbf{N}, \bar{\mathbf{N}})$ representation of $\mathrm{U}(N)_k \times \mathrm{U}(N)_k$ gauge groups and the other two, $B_{1,2}$, transform in the $(\bar{\mathbf{N}}, \mathbf{N})$ representation. The superpotential is given by

$$\mathcal{W} = \lambda\left[\mathrm{Tr}\,\Phi_1(A_1 B_1 + A_2 B_2) + \mathrm{Tr}\,\Phi_2(B_1 A_1 + B_2 A_2)\right] \,. \tag{10}$$

This superpotential leads to all chiral superfields having R-charge 2/3. Therefore we have a total of 10 candidate marginal operators that are obtained as level 2 descendants of the following gauge invariant chiral operators of R-charge 2

$$
\begin{array}{ccccc}
\mathrm{Tr}\,\Phi_1 A_1 B_1\,, & \mathrm{Tr}\,\Phi_1 A_1 B_2\,, & \mathrm{Tr}\,\Phi_1 A_2 B_1\,, & \mathrm{Tr}\,\Phi_1 A_2 B_2\,, & \mathrm{Tr}\,\Phi_1^3\,, \\
\mathrm{Tr}\,\Phi_2 B_1 A_1\,, & \mathrm{Tr}\,\Phi_2 B_1 A_2\,, & \mathrm{Tr}\,\Phi_2 B_2 A_1\,, & \mathrm{Tr}\,\Phi_2 B_2 A_2\,, & \mathrm{Tr}\,\Phi_2^3\,.
\end{array}
\tag{11}
$$

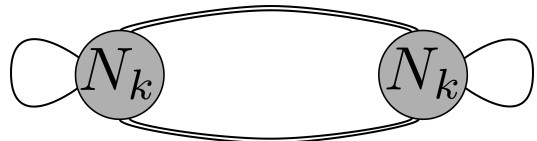

Figure 2: Quiver representation of the 4d $\mathbf{Z}_2$ orbifold of the 4d $\mathcal{N} = 4$ SYM theory interpreted as a 3d $\mathcal{N} = 2$ quiver with equal Chern-Simons levels.

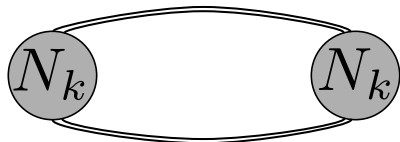

Figure 3: Quiver representation of the 4d $\mathcal{N} = 1$ Klebanov-Witten theory interpreted as a 3d $\mathcal{N} = 2$ quiver with equal Chern-Simons levels.

To count the exactly marginal operators in this model we first discuss the global symmetries in the theory without a superpotential. One linear combination of the two U(1) factors in the $U(N)_k \times U(N)_k$ gauge group acts trivially and will not play a role. The other linear combination results in a topological $U(1)_T$ global symmetry.[5] In addition the theory with no superpotential has a $U(2) \times U(2)$ global flavor symmetry. We therefore conclude that the complex dimension of the conformal manifold is

$$\dim_{\mathbf{C}}(\mathcal{M}_c) = 10 - \dim(U(2) \times U(2) \times U(1)_T) = 10 - 9 = 1 \,. \tag{12}$$

Note that this result agrees with the intuition from the 4d cousin of this SCFT. As discussed in [28] the 4d theory has a conformal manifold of complex dimension 3 where 2 of the exactly marginal couplings are given by the complexified gauge couplings that are not present in the 3d SCFT. All candidate marginal operators in (11) are invariant under the $U(1)_T$ global symmetry which acts with equal and opposite charges on the $A_{1,2}$ and $B_{1,2}$ chiral superfields. Therefore, the whole conformal manifold enjoys this global symmetry in addition to the omnipresent U(1) R-symmetry. The $S^3$ free energy of this SCFT can be readily computed using the results in [16] to find

$$\operatorname{Re} \mathcal{F}_{2,\mathrm{adj}} = \frac{2^{4/3} 3^{1/6}}{5} \pi k^{1/3} N^{5/3} \,. \tag{13}$$

We can deform this family of SCFT by adding a mass term for the chiral adjoint superfields to the superpotential, i.e. we add the following relevant terms to the superpotential in (10)

$$\mathcal{W} = m \left[ \operatorname{Tr} \Phi_1^2 - \operatorname{Tr} \Phi_2^2 \right] \,. \tag{14}$$

In the deep IR we can integrate out the chiral superfields $\Phi_{1,2}$ and arrive at a 3d $\mathcal{N} = 2$ SCFT which is a direct analog of the 4d $\mathcal{N} = 1$ Klebanov-Witten SCFT studied in [29], see Figure 3. The superpotential for this model is given by

$$\mathcal{W}_{\mathrm{KW}} = \epsilon^{ij} \epsilon^{kl} \operatorname{Tr}(A_i B_k A_j B_l) \,. \tag{15}$$

The global symmetry of this theory is $SU(2)_A \times SU(2)_B \times U(1)_R \times U(1)_T$. All four chiral superfields are charged under the superconformal R-symmetry $U(1)_R$ with charge $q_R = 1/2$ and have conformal dimension $\Delta = 1/2$. The fields $A_i$ and $B_j$ transform as doublets under the $SU(2)_A$ and $SU(2)_B$ flavor symmetries. Finally, the topological $U(1)_T$ global symmetry acts with charge $+1$ on the $A_i$ and $-1$ on the $B_j$.

All single-trace marginal operators have dimension $\Delta = 2$ and are of the form $\lambda^{ij,kl} \operatorname{Tr}(A_i B_k A_j B_l)$. Since the $A_i$ and $B_j$ transform respectively in the $(\mathbf{2}, \mathbf{1})$ and $(\mathbf{1}, \mathbf{2})$ representations of $SU(2)_A \times SU(2)_B$, a generic quartic term transforms as

$$(\mathbf{2}, \mathbf{1}) \otimes (\mathbf{1}, \mathbf{2}) \otimes (\mathbf{2}, \mathbf{1}) \otimes (\mathbf{1}, \mathbf{2}) = (\mathbf{1}, \mathbf{1}) \oplus (\mathbf{3}, \mathbf{1}) \oplus (\mathbf{1}, \mathbf{3}) \oplus (\mathbf{3}, \mathbf{3}) \,. \tag{16}$$

The trace annihilates the terms $(\mathbf{1}, \mathbf{3})$ and $(\mathbf{3}, \mathbf{1})$ so the most general single trace superpotential deformation is given by

$$\delta \mathcal{W} = \lambda^{(ij)(kl)} \operatorname{Tr}(A_i B_k A_j B_l) \,, \tag{17}$$

---

[5]One can alternatively consider this theory with an $SU(N)_k \times SU(N)_k$ gauge group. In this case the $U(1)_T$ plays the role of a baryonic global symmetry.

where $\lambda^{(ij)(kl)}$ is symmetric in the indices $i, j$ and $k, l$. Therefore, after accounting for the chiral ring relations coming from the superpotential in (15), there are 10 candidate marginal superpotential terms, 1 that preserves $SU(2)_A \times SU(2)_B$ and 9 that violate it. The complex dimension of the conformal manifold $\mathcal{M}_c$ of this three-dimensional $\mathcal{N} = 2$ SCFT is therefore

$$\dim_{\mathbf{C}}(\mathcal{M}_c) = 10 - \dim(SU(2)_A \times SU(2)_B \times U(1)_T) = 3. \tag{18}$$

Again, this agrees with the expectation from four dimensions where one finds a five-dimensional conformal manifold where the two extra exactly marginal deformations can be accounted for by the complexified gauge couplings [28]. The three exactly marginal deformations are induced by the operators

$$\mathrm{Tr}\,(A_1 B_1 A_2 B_2 + A_1 B_2 A_2 B_1), \tag{19}$$

$$\mathrm{Tr}\,(A_1 B_1 A_1 B_1 + A_2 B_2 A_2 B_2), \tag{20}$$

$$\mathrm{Tr}\,(A_1 B_2 A_1 B_2 + A_2 B_1 A_2 B_1). \tag{21}$$

The full superpotential with generic couplings for the exactly marginal deformations can thus be written as

$$
\begin{aligned}
\mathcal{W} =& \mathrm{Tr}\,(A_1 B_1 A_2 B_2 - A_1 B_2 A_2 B_1) \\
&+ \lambda_{\mathrm{PW}}\,\mathrm{Tr}\,(A_1 B_1 A_2 B_2 + A_1 B_2 A_2 B_1 - A_1 B_2 A_1 B_2 - A_2 B_1 A_2 B_1) \\
&+ \lambda_{\beta}\,\mathrm{Tr}\,(A_1 B_1 A_2 B_2 + A_1 B_2 A_2 B_1) \\
&+ \lambda_2\,\mathrm{Tr}\,(A_1 B_1 A_1 B_1 + A_2 B_2 A_2 B_2),
\end{aligned}
\tag{22}
$$

where the couplings are labeled following the notation for the four-dimensional parent theory, see [28]. The term proportional to $\lambda_{\mathrm{PW}}$ preserves an $SU(2) \times U(1)_T$ global symmetry analogous to the four-dimensional Pilch-Warner deformation [30]. The term proportional to $\lambda_{\beta}$ is the analog of the $\beta$-deformation in (3) and preserves the $U(1)^3$ Cartan subalgebra of the flavor symmetry. Finally, turning on the last coupling $\lambda_2$ preserves only the $U(1)_T$ global symmetry. Putting $\lambda_{\mathrm{PW}} = \lambda_2 = 0$ the superpotential (15) is deformed to

$$\mathcal{W}_{\beta} = \mathrm{Tr}\,(e^{i\pi\beta} A_1 B_1 A_2 B_2 - e^{-i\pi\beta} A_1 B_2 A_2 B_1), \tag{23}$$

which makes the analogy with (3) manifest. The $S^3$ free energy of this SCFT can again can be computed in the large $N$ limit using supersymmetric localization with the result

$$\mathrm{Re}\,\mathcal{F}_2 = \frac{3^{13/6}}{20}\pi k^{1/3} N^{5/3}. \tag{24}$$

### Necklace quivers

We now study a natural generalization of the constructions above given by a 4d parent theory obtained by a $\mathbf{Z}_b$ orbifold of the $\mathcal{N} = 4$ SYM theory. These 4d $\mathcal{N} = 2$ theories are succinctly summarized by the necklace quiver in Figure 4.[6] The field content of these theories consists of $b$ vector multiplets, $b$ adjoint chirals, and $b$ chiral multiplets in the representations $(1, \cdots, N_{(i)}, \overline{N}_{(i+1)}, \cdots, 1)$ and $b$ chiral multiplets in the representations $(1, \cdots, \overline{N}_{(i)}, N_{(i+1)}, \cdots, 1)$ where $N_{(i)}$ and $\overline{N}_{(i)}$ denote the fundamental representation of the $i^{\mathrm{th}}$ gauge group. We call these fields $\Phi_i$, $A_i$ and $\tilde{A}_i$, respectively. The quiver is given in Figure 4 and as usual we add a Chern-Simons level $k$ to each of the gauge nodes. The superpotential for this model is the same as the one for the 4d $\mathcal{N} = 2$ parent theory and is given by[7]

$$\mathcal{W} = \sum_{i=1}^{b} \lambda_i \mathrm{Tr}\,\left(\Phi_i A_i \tilde{A}_i\right) + \sum_{i=1}^{b} \tilde{\lambda}_i \mathrm{Tr}\,\left(\Phi_i \tilde{A}_{i+1} A_{i+1}\right). \tag{25}$$

---

[6]These theories are the same as the $L^{0b0}$ SCFTs studied in [31].

[7]Due to the cyclicity of the quiver we identify $b + 1 \equiv 1$.

Clearly all superfields have charges 2/3 under the U(1) R-symmetry. In addition the model

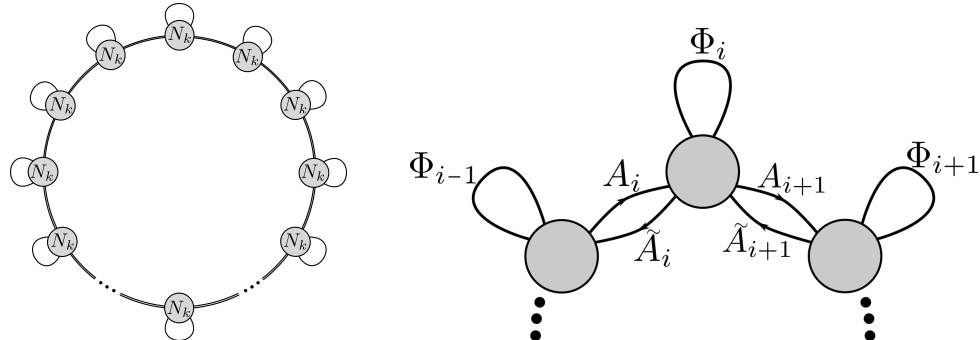

Figure 4: Left: A necklace quiver gauge theory with $b$ nodes obtained by performing a $\mathbf{Z}_b$ orbifold of $\mathcal{N} = 4$ SYM. Right: The quiver representation of the theory consists of $b$ gauge nodes, each with Chern-Simons level $k$, connected with a bifundamental hypermultiplet. In addition a chiral adjoint is attached to every node.

with no superpotential enjoys a large flavor symmetry. To every pair of bi-fundamental chiral multiplets we can assign a U(1) action with generators $J_i$ under which the fields are charged as $J_i(A_j) = J_i(\tilde{A}_j) = \delta_{ij}$ and $J_i(\Phi_j) = 0$. Similarly, to every chiral adjoint there is an associated U(1) action with generators $F_i$ under which the fields are charged as $F_i(A_j) = F_i(\tilde{A}_j) = 0$ and $F_i(\Phi_j) = \delta_{ij}$. Using these charges one can then check that the linear combinations $J_{i+1} - J_i$ and $J_i + J_{i-1} - 2F_i$ lead to U(1)$^{2b}$ flavor symmetries. In addition to these symmetries there are $b - 1$ U(1) "baryonic" symmetries that act with equal and opposite charges on each pair $(A_i, \tilde{A}_i)$.[8]

There are a total of $3b$ candidate marginal operators in the theory with no superpotential which are obtained as descendants of the following chiral operators of R-charge 2.

$$\text{Tr}\left(\Phi_i A_i \tilde{A}_i\right), \qquad \text{Tr}\left(\Phi_i \tilde{A}_{i+1} A_{i+1}\right), \qquad \text{Tr}\left(\Phi_i^3\right), \qquad i = 1, 2, 3, \ldots, b. \tag{26}$$

Following [17–19] we find that the dimension of the conformal manifold is given by

$$\dim_{\mathbf{C}} \mathcal{M}_c = 3b - \dim(\text{U}(1)^{2b} \times \text{U}(1)^{b-1}) = 3b - 2b - (b-1) = 1. \tag{27}$$

This result again agrees with the intuition obtained from the 4d parent theory where it was argued in [32] that the operators $\text{Tr}\left(\Phi_i^3\right)$ are marginally irrelevant and only a single linear combination of the operators $\text{Tr}\left(\Phi_i A_i \tilde{A}_i\right)$ and $\text{Tr}\left(\Phi_i \tilde{A}_{i+1} A_{i+1}\right)$ is exactly marginal. This exactly marginal deformation is exactly the analog of the $\beta$-deformation of $\mathcal{N} = 4$ SYM which survives the orbifold action and preserves a U(1)$^2$ subgroup of the flavor symmetry.

The $S^3$ free energy of the necklace quiver with adjoint chirals can be computed using the results in [16] and reads

$$\text{Re}\,\mathcal{F}_{b,\text{adj}} = \frac{2^{1/3} 3^{1/6}}{5} b \, \pi k^{1/3} N^{5/3}. \tag{28}$$

We can generalize the construction above by adding a relevant mass term to the superpotential of the form

$$\sum_i \frac{m_i}{2} \text{Tr}(\Phi_i^2). \tag{29}$$

---

[8]In the quiver with U($N$) gauge groups, that we study here, at every node these "baryonic" symmetries are actually part of gauge group however each of them leads to global topological symmetry acting on monopole operators, similar to the the U(1)$_{\text{T}}$ discussed below (11). One can alternatively consider a quiver gauge theory with SU($N$) gauge groups at all nodes in which case these are global baryonic symmetries.

This type of mass deformations of the necklace quivers are well known in the 4d SCFT context, see for instance [33]. In the IR one can integrate out the adjoint chiral superfields and arrive at a necklace quiver of the same form as in Figure 4 but with no "adjoint lines" attached to each node. The bifundamental chiral multiplets are still present in the IR and the theory has a quartic superpotential similar to the one of the KW theory. Note that for quivers with even number of nodes, i.e. $b = 2a$, the IR theory is the same as the ones described by the $L^{aaa}$ quiver gauge theories, see [31]. Since the superpotential is quartic in the bifundamental chiral superfields all of them have equal R-charge equal to $1/2$. The global symmetry associated with the adjoint chiral superfields is not present any more and therefore these models (with no superpotential) have only $U(1)^b$ flavor symmetry along with the $U(1)^{b-1}$ "baryonic" symmetry discussed above. The candidate marginal operators in these models are given by the

$$\text{Tr}\left(A_i \tilde{A}_i A_{i+1} \tilde{A}_{i+1}\right), \qquad \text{Tr}\left(A_i \tilde{A}_{i+1} A_{i+1} \tilde{A}_i\right), \qquad i = 1, 2, 3, \ldots, b. \tag{30}$$

We can now once again utilize the general result in [17–19] to conclude that the dimension of the conformal manifold for these quiver SCFTs is given by

$$\dim_{\mathbb{C}} \mathcal{M}_c = 2b - \dim(U(1)^b \times U(1)^{b-1}) = 2b - b - (b-1) = 1. \tag{31}$$

We can once again compute the $S^3$ free energy of these SCFTs to find

$$\text{Re}\,\mathcal{F}_b = \frac{3^{13/6}}{5 \cdot 2^3} b\, \pi k^{1/3} N^{5/3}. \tag{32}$$

## 2.2 Comments on RG flows

In all models discussed above we have the option to deform the theory with adjoint chiral superfields by a superpotential mass term and thus flow to a new IR SCFT. These types of RG flows are well-known in the context of 4d SCFT where they were shown to exhibit universal properties in [34]. The discussion above naturally suggests that there is a similar universality in the class of Chern-Simons-matter theories of interest here. In particular for all RG flows triggered by masses for adjoint chiral superfields above we find the same ratio of $S^3$ free energies between the IR and the UV SCFT, namely

$$\frac{\text{Re}\,\mathcal{F}_{\text{IR}}}{\text{Re}\,\mathcal{F}_{\text{UV}}} = \left(\frac{27}{32}\right)^{2/3} \approx 0.892913. \tag{33}$$

This result is of course compatible with the F-theorem and can be viewed as a 3d $\mathcal{N} = 2$ analogue of the universal ratio between IR and UV conformal anomalies for the 4d SCFTs discussed in [34]. We should stress that the result in (33) is valid only to leading order in the large $N$ expansion of the gauge theory free energy. It will be most interesting to understand whether this universality persists to subleading order and what is precisely the class of 3d $\mathcal{N} = 2$ SCFTs to which it applies.

## 2.3 A digression on ABJM

Before discussing the supergravity background dual to the above-mentioned SCFTs, let us briefly discuss the ABJM theory and a closely related SCFT obtained by a mass deformation. The ABJM theory is a double Chern-Simons-matter theory [35] very similar to the three-dimensional KW SCFT discussed above. The gauge group is $U(N)_k \times U(N)_{-k}$ where in contrast to the above, the Chern-Simons levels are now given by $k$ and $-k$. Since the two levels are equal and opposite this theory preserves parity. The field content, superpotential and global symmetries of this theory, however, are identical to those of the three-dimensional KW theory.

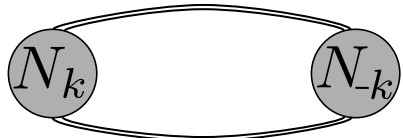

Figure 5: Quiver representation of the ABJM theory.

This theory is conformal and describes the dynamics of $N$ M2-branes probing a $\mathbf{C}^4/\mathbf{Z}_k$ singularity in M-theory. The $\mathbf{Z}_k$ acts as an $e^{2\pi i/k}$ rotation on the four complex planes transverse to the stack of branes. There are various generalizations of this setup to $U(N)_{k_1} \times U(M)_{k_2}$ Chern-Simons-matter theory with $N \neq M$, see [36], and $k_1 + k_2 \neq 0$, see [14], which we do not discuss further here. When $k = 1, 2$ the ABJM theory preserves $\mathcal{N} = 8$ supersymmetry. For $k > 2$ on the other hand, supersymmetry is partially broken to $\mathcal{N} = 6$. These two cases have important differences and we discuss them separately below. Our goal is to understand the space of exactly marginal deformation of this class of SCFTs which preserve $\mathcal{N} = 2$ supersymmetry.[9]

### 2.3.1 $k > 2$

The ABJM theory with $k > 2$ has $\mathcal{N} = 6$ superconformal symmetry and an $SO(6) \simeq SU(4)$ R-symmetry. This is not manifest in superspace but can be deduced when the Lagrangian of the theory is written in components [35]. The $U(1)_R \times SU(2)_A \times SU(2)_B$ is a subgroup of the $SU(4)$ R-symmetry. The $A$ and $B$ fields can be organized in representation of $SU(4) \times U(1)_b$ (see for example [38]) as follows

$$
\begin{aligned}
(A_1, A_2, B_1^\dagger, B_2^\dagger), \qquad &\text{transforms in the} \qquad &\mathbf{4}_1\,, \\
(A_1^\dagger, A_2^\dagger, B_1, B_2), \qquad &\text{transforms in the} \qquad &\overline{\mathbf{4}}_{-1}\,.
\end{aligned}
\tag{34}
$$

Since the R-charges and dimensions of the chiral primaries as well as the chiral ring relations are exactly the same as in the three-dimensional Klebanov-Witten SCFT, we find the same exactly marginal operators and consequently the complex dimension of the conformal manifold is given by $\dim_{\mathbf{C}}(\mathcal{M}_c^{\text{ABJM}_{k>2}}) = 3$. This result was also found in [9].

### 2.3.2 $k = 1, 2$

For $k = 1, 2$, there is an important new ingredient in the discussion, namely monopole operators [39, 40]. The monopole operators, $T^{(q)}$, are not constructed out of Lagrangian building blocks but can be thought of as point-like insertions which turn on $q \in \mathbf{Z}$ units of flux (through the $S^2$ surrounding the point) for the topological current $*_3 \text{Tr}(F + \widetilde{F})$. For the $k = 1$ theory the operators $T^{(1)}$ and $T^{(-1)}$ reside in respectively the $(\mathbf{N}, \overline{\mathbf{N}})$ and $(\overline{\mathbf{N}}, \mathbf{N})$ representations of the gauge groups. One can consistently choose the charge under $U(1)_R$ of these operators to be 0. Note that in order to obtain chiral primary operators one has to form a gauge invariant product of $A_{1,2}$, $B_{1,2}$, $T^{(1)}$, and $T^{(-1)}$. For example, $\text{Tr}(T^{(1)} A_1 T^{(1)} A_1)$ is a chiral primary operator of dimension and R-charge $\Delta = q_R = 1$.[10] The existence of these monopole operators is ultimately what is responsible for the enhancement of supersymmetry to $\mathcal{N} = 8$ and the R-symmetry to $SO(8)$ [41, 42]. For the $k = 2$ theory the relevant operators to consider are $T^{(2)}$ and $T^{(-2)}$ which transform in $(\mathbf{N} \otimes \mathbf{N}, \overline{\mathbf{N}} \otimes \overline{\mathbf{N}})$ representation of the gauge group and lead to chiral primary operators of the form $\text{Tr}(A_1 T^{(2)} A_1)$. To avoid repetition we will focus on discussing the $k = 1$

---

[9]The general results in [37] imply that there are no exactly marginal deformations in 3d SCFTs which preserve more than $\mathcal{N} = 2$ supersymmetry.

[10]Operators of the type $\text{Tr}(T^{(1)} A_1)$ or $\text{Tr}(T^{(-1)} B_2)$ have $\Delta = q_R = 1/2$ and thus saturate the unitarity bound in 3d. There are 8 such operators which is exactly the right number to account for the decoupled Abelian M2-brane theory which is free and describes the "center of mass" mode of the stack of $N$ M2-branes.

theory below, see Section 3.2 of [43] for more explicit details on the monopole operators and corresponding gauge invariant operators for the $k = 2$ model.

For the purposes of studying the conformal manifold we can formally define 4 chiral primary operators

$$\Phi_1 = T^{(1)}A_1, \qquad \Phi_2 = T^{(1)}A_2, \qquad \Phi_3 = T^{(-1)}B_1, \qquad \Phi_4 = T^{(-1)}B_2, \qquad (35)$$

where the fields $\Phi_i$ belong to the $\mathbf{4}_{1/2}$ representation of the $SU(4)_F \times U(1)_R$ subgroup of $SO(8)$. Here $SU(4)_F$ is the commutant of $U(1)_R$ inside $SO(8)$, in other words the manifest flavor symmetry $SU(2)_A \times SU(2)_B \times U(1)_b$ is a subgroup of $SU(4)_F$. The candidate marginal operators are descendants of quartic operators with R-charge 2 of the form

$$\mathcal{W} \sim h^{ijkl} \text{Tr}\left(\Phi_i \Phi_j \Phi_k \Phi_l\right). \qquad (36)$$

The presence of the superpotential in (15) gives rise to several nontrivial chiral ring relations [44]. These relations arise from the equation $\partial_{A_{1,2}} \mathcal{W} = \partial_{B_{1,2}} \mathcal{W} \sim 0$ where $\sim 0$ means zero up to non-chiral primaries. In particular, this results in the following 4 relations

$$\begin{aligned} B_1 A_2 B_2 \sim B_2 A_2 B_1, \qquad B_2 A_1 B_1 \sim B_1 A_1 B_2, \\ A_2 B_2 A_1 \sim A_1 B_2 A_2, \qquad A_1 B_1 A_2 \sim A_2 B_1 A_1. \end{aligned} \qquad (37)$$

These relations should be treated formally and should only be used in gauge invariant combinations and thus could be sprinkled with monopole operators. Due to the presence of these relations, we can take the constants $h^{ijkl}$ in (36) to be the components of a completely symmetric tensor.

The flavor symmetry is $SU(4)_F$ and hence the dimension of the conformal manifold is

$$\dim_{\mathbf{C}}(\mathcal{M}_c^{\text{ABJM}_{k=1,2}}) = 35 - \dim(SU(4)_F) = 20, \qquad (38)$$

in agreement with the counting in [18]. As explained in [18] the 35 naive complex marginal deformations are in the $\mathbf{35} \oplus \overline{\mathbf{35}}$ representation of $SU(4)_F$. There is a special one-dimensional (complex) submanifold of the conformal manifold on which the $U(1)^3$ Cartan subgroup of $SU(4)_F$ is preserved. It is spanned by the two singlets arising from the decomposition of the $\mathbf{35}$ and $\overline{\mathbf{35}}$ under $U(1)^3$. This one-dimensional conformal sub-manifold should be the direct analog of the $\beta$ deformation of the $\mathcal{N} = 4$ SYM theory and it is natural to wonder what is its gravitational dual description. Indeed, a deformation of the $AdS_4 \times S^7$ solution of 11d supergravity with precisely this isometry was described in [11] however it is controlled by a single real parameter $\hat{\gamma}$. We are not aware of an $AdS_4$ supergravity solution which realizes the more general complex marginal deformation preserving $U(1)^3 \times U(1)_R$.

### 2.3.3 mABJM

The ABJM theory with $k = 1$ can be deformed by the superpotential term[11]

$$\mathcal{W}_m = m^2 \text{Tr}(T^{(1)} A_1 T^{(1)} A_1). \qquad (39)$$

This is a relevant operator of conformal dimension and R-charge $\Delta = q_R = 1$ which triggers an RG flow. The resulting IR dynamics is controlled by another strongly interacting 3d $\mathcal{N} = 2$ SCFT which we refer to as mABJM. This 3d theory was introduced soon after the ABJM theory in [45] (see also [24, 43]) and the gravity dual in four-dimensional gauged supergravity was constructed in the early days of gauged supergravity by Warner [46]. The corresponding $AdS_4$

---

[11]This deformation is also possible in the theory with $k = 2$ where the superpotential is $\mathcal{W}_m = m^2 \text{Tr}(A_1 T^{(2)} A_1)$.

solution in 11d was found in [47]. The space of exactly marginal deformations of this theory as well as its gravitational dual however have not been studied in the literature.

Formally, we can think of the deformation (39) as $\text{Tr}(\Phi_1^2)$. This modifies the R-charge assignments for $\Phi_{2,3,4}$ such that they now have R-charge $q_r = \frac{1}{3}$. Another way to say this is that the IR superconformal R-symmetry, $U(1)_r$, is obtained by breaking $SU(4)_F \times U(1)_R$ to $SU(3)_F \times U(1)_f \times U(1)_R$ and then taking $U(1)_r$ to be the diagonal of $U(1)_f \times U(1)_R$. The candidate supersymmetric marginal deformations are then sextic operators built out of the $\Phi_{2,3,4}$. These are deformations of the superpotential of the type

$$W \sim \lambda^{i_1 i_2 i_3 i_4 i_5 i_6} \Phi_{i_1} \Phi_{i_2} \Phi_{i_3} \Phi_{i_4} \Phi_{i_5} \Phi_{i_6}, \qquad i_k = 2, 3, 4. \tag{40}$$

Using the chiral ring relations that follow from combining the superpotentials in (15) and (39) one finds that the tensor $\lambda^{i_1 i_2 i_3 i_4 i_5 i_6}$ has to be completely symmetric in its indices, i.e. it should be in the **28** representation of $SU(3)_F$. We thus arrive at the conclusion that the dimension of the conformal manifold of the mABJM theory should be

$$\dim_{\mathbf{C}}(\mathcal{M}_c^{\text{mABJM}}) = 28 - \dim(SU(3)_F) = 20. \tag{41}$$

Curiously, the dimensions of the conformal manifolds of the $k = 1, 2$ ABJM and mABJM SCFTs are exactly the same. We can again ask how many operators are singlets of the Cartan $U(1)^2$ of $SU(3)_F$. We have one from the **28** and one from the $\overline{\mathbf{28}}$. This indicates that the conformal manifold of the mABJM theory contains a one-complex-dimensional submanifold on which the $U(1)^2 \times U(1)_R$ symmetry is preserved. This should be manifested holographically as a one-parameter family of deformations of the CPW solution [47] which preserves all supersymmetries as well as the isometries of $AdS_4$. It will be most interesting to construct this solution explicitly.

We end our digression with a short discussion of the $S^3$ free energies of the ABJM and mABJM theory, see [24]. They can be computed by supersymmetric localization and read

$$F_{S^3}^{\text{ABJM}} = \frac{\sqrt{2}\pi}{3} N^{3/2}, \qquad F_{S^3}^{\text{mABJM}} = \frac{4\sqrt{2}\pi}{9\sqrt{3}} N^{3/2}. \tag{42}$$

The ratio of these two quantities is equal to $4/3^{3/2}$ and therefore the RG flow connecting the ABJM and mABJM theory does not belong to the same "universality class" as the one described around (33).

## 3 Supergravity

We now switch gears to discuss the holographic dual description of the 3d $\mathcal{N} = 2$ SCFTs presented above together with some of their marginal deformations. We focus on $AdS_4$ solutions of type IIA supergravity with non-vanishing Romans mass since the SCFTs of interest break parity and have non-vanishing sum of the Chern-Simons levels. We start by introducing a set of "seed solutions" corresponding to the undeformed SCFTs. We then introduce the holographic dual of a marginal deformation by performing the TsT transformation discussed in [11]. As discussed in the previous section, the 3d SCFTs are related to certain 4d parent SCFTs. In the same spirit, the $AdS_4$ solutions we discuss are related to parent $AdS_5$ solutions. The 4d parent $\mathcal{N} = 1$ quiver gauge theory can be engineered as the worldvolume theory of a stack of $N$ D3-branes on the background $\mathbf{R}^{1,3} \times X$. The dual supergravity background then describes the back-reaction of this stack of D3-branes and is of the form $AdS_5 \times Y^5$ where $Y^5$ is a five-dimensional Sasaki-Einstein manifold which is given as the link of the Calabi-Yau space $X$. To this solution we can associate a solution in massive type IIA supergravity which has

the form $AdS_4 \times M_6$, where $M_6 \simeq \mathcal{S}Y^5$ is the suspension of $Y^5$ [15,16]. Indeed, this solution then describes a stack of $N$ D2-branes probing $\mathbf{R} \times X$. A first example of this construction was discussed in [15], where the dual to the one node quiver theory in Figure 1 was constructed. This was subsequently generalized to a larger class of quiver gauge theories in [16].

## 3.1 Seed solutions

The massive type IIA supergravity backgrounds of interest take the general form [15,16]:

$$
\begin{aligned}
\mathrm{d}s_{10}^2 &= \frac{\sqrt{2 + \cos^2 \alpha}}{3(g^5 m)^{1/3}} \left[ \mathrm{d}s_{AdS_4}^2 + \frac{3}{2}\mathrm{d}\alpha^2 + \frac{3\sin^2 \alpha}{1 + \cos^2 \alpha}\mathrm{d}s_{\mathcal{B}}^2 + \frac{9\sin^2 \alpha}{2(2 + \cos^2 \alpha)}\eta^2 \right], \\
\mathrm{e}^\Phi &= \left(\frac{g}{m}\right)^{5/6} \frac{(2 + \cos^2 \alpha)^{3/4}}{1 + \cos^2 \alpha}, \\
H_3 &= \frac{2}{(g^5 m)^{1/3}} \frac{\sin^3 \alpha}{(1 + \cos^2 \alpha)^2} J \wedge \mathrm{d}\alpha, \\
F_0 &= m, \\
F_2 &= -\left(\frac{m^2}{g^5}\right)^{1/3} \left( \frac{\sin^2 \alpha \cos \alpha}{(1 + \cos^2 \alpha)(2 + \cos^2 \alpha)} J + \frac{3(2 - \cos^2 \alpha)\sin \alpha}{2(2 + \cos^2 \alpha)^2}\mathrm{d}\alpha \wedge \eta \right), \\
F_4 &= \left(\frac{m}{g^{10}}\right)^{1/3} \left( \frac{1}{\sqrt{3}}\mathrm{vol}_{AdS_4} + \frac{(2 + 3\cos^2 \alpha)\sin^4 \alpha}{2(1 + \cos^2 \alpha)^2}J \wedge J \right. \\
&\quad \left. + \frac{3(4 + \cos^2 \alpha)\sin^3 \alpha \cos \alpha}{2(1 + \cos^2 \alpha)(2 + \cos^2 \alpha)}J \wedge \mathrm{d}\alpha \wedge \eta \right),
\end{aligned}
\tag{43}
$$

where $\mathrm{d}s_{10}^2$ is the ten-dimensional metric in Einstein frame. The AdS metric, $\mathrm{d}s_{AdS_4}^2$ has unit AdS length and volume form $\mathrm{vol}_4$. $\mathcal{B}$ is a four-dimensional Kähler-Einstein base manifold with Ricci tensor $R_{\mu\nu} = 6g_{\mu\nu}$, $J$ is the corresponding transverse Kähler form and $\eta = \mathrm{d}\psi + \sigma$ is a suitable contact one-form such that $\mathrm{d}\eta = 2J$. We refer to Appendix A for our supergravity conventions. The precise form of the gauge potentials for the seed solutions can be found in Appendix B. As mentioned above, we can relate these massive type IIA backgrounds to type IIB backgrounds, corresponding to the four-dimensional parent theory. The four-dimensional parent field theory has an $AdS_5 \times Y^5$ dual, where $Y^5$ is a Sasaki-Einstein manifold with base $\mathcal{B}$, i.e. with Sasaki-Einstein metric

$$
\mathrm{d}s_{Y^5}^2 = \eta^2 + \mathrm{d}s_{\mathcal{B}}^2 .
\tag{44}
$$

Topologically, the near horizon geometries of our seed solutions are warped products of the form $AdS_4 \times \mathcal{S}Y^5$ where $\mathcal{S}Y^5$ denotes the suspension of the Sasaki-Einstein space $Y^5$. Due to the additional fluxes, the suspended geometry is deformed but the topology remains the same. The coordinate $\alpha$ ranges from 0 to $\pi$ and, for generic $Y^5$, one encounters isolated Calabi-Yau conical singularities at the endpoints of this interval, inherited from the singularity of $X$. It is natural to expect stringy degrees of freedom to be supported at the singularities, however, these appear to not contribute to some physical observables at leading order in the large $N$ expansion. We provide evidence for this claim by computing the $S^3$ free energy of both the seed and TsT deformed solution and comparing it with the leading order field theory answer.

We are interested in deformations of the $AdS_4$ solution above that describe the exactly marginal couplings parameterizing the conformal manifold discussed in Section 2. We do not know how to construct the general supergravity solution dual to these marginal deformations and will only study a particular one-dimensional real submanifold of the conformal manifold described by the TsT transformation of Lunin-Maldacena [11]. To implement this series of duality transformations on the supergravity solution above and preserve supersymmetry we

need to choose spaces $\mathcal{B}$ that have at least $U(1)^2$ invariance, i.e. we need $\mathcal{B}$ to be a toric Kähler-Einstein 4-manifold with positive curvature. Combining results from [48] and [49] we find that the only regular Kähler-Einstein bases satisfying these conditions are $\mathbf{CP}^2$, $S^2 \times S^2$ and the third del Pezzo surface $dP_3$.

For $\mathcal{B} = \mathbf{CP}^2$ the dual 3d SCFT is the single node theory discussed in Section 2, see Figure 1. In this case the associated Sasaki-Einstein manifold is $Y^5 = S^5$ and the topology of the internal space is $S^6$. The full solution exhibits an $SU(3)_F \times U(1)_R$ symmetry in agreement with the field theory analysis. For convenience, we use the following toric metric on $\mathbf{CP}^2$,

$$
\begin{aligned}
ds^2_{\mathbf{CP}^2} &= d\alpha_0^2 + \frac{\sin^2 \alpha_0}{4} \left( \sigma_1^2 + \sigma_2^2 + \cos^2 \alpha_0 \, \sigma_3^2 \right), \\
\sigma_{\mathbf{CP}^2} &= \frac{\sin^2 \alpha_0}{2} \sigma_3,
\end{aligned}
\tag{45}
$$

where $\sigma_{\mathbf{CP}^2}$ is related to the one-form $\eta$ as described below (43) and $\sigma_i$ are the $SU(2)$ left-invariant one-forms

$$
\begin{aligned}
\sigma_1 &= \cos \alpha_3 \, d\alpha_1 + \sin \alpha_1 \sin \alpha_3 \, d\alpha_2, \\
\sigma_2 &= \sin \alpha_3 \, d\alpha_1 - \sin \alpha_1 \cos \alpha_3 \, d\alpha_2, \\
\sigma_3 &= d\alpha_3 + \cos \alpha_1 \, d\alpha_2.
\end{aligned}
\tag{46}
$$

For $\mathcal{B} = S^2 \times S^2$ the associated Sasaki-Einstein Manifold is $Y^5 = T^{1,1} \simeq S^2 \times S^3$ and the global symmetry is given by $SU(2)_F^2 \times U(1)_b \times U(1)_R$. The two $SU(2)$ symmetries are associated to the two $\mathbf{CP}^1$ factors of $T^{1,1}$ and the $U(1)_R$ is associated to the fiber. The baryonic symmetry on the other hand is not associated to the isometries but to the presence of the nontrivial three-cycle in the topology of $T^{1,1}$. The dual field theory is given by a $U(N)_k \times U(N)_k$ Chern-Simons gauge theory with four bifundamental chirals corresponding to the three-dimensional $\mathcal{N} = 2$ Klebanov-Witten SCFT discussed in Section 2, see Figure 3. The explicit form of the metric and one-form needed to specify the solution in (43) are given by

$$
\begin{aligned}
ds^2_{\mathcal{B}} &\to ds^2_{S^2 \times S^2} = \frac{1}{6}(d\alpha_0^2 + d\alpha_1^2 + \sin^2 \alpha_0 d\alpha_2^2 + \sin^2 \alpha_1 d\alpha_3^2), \\
\sigma &\to \sigma_{S^2 \times S^2} = \frac{1}{3}(\cos \alpha_0 d\alpha_2 + \cos \alpha_1 d\alpha_3).
\end{aligned}
\tag{47}
$$

We have explicitly checked that the equations of motion of the 10d supergravity theory are indeed satisfied.

In order to find the supergravity description of the more general necklace quivers discussed in Section 2 we need to relax the regularity condition on the base $\mathcal{B}$ and allow for orbifold singularities. In particular the Sasaki-Einstein space relevant for the description of the the necklace quivers with adjoint chirals and $b$ nodes is $Y^5 = S^5/\mathbf{Z}_b$ which is the same as the $L^{0b0}$ manifold. For the necklace quivers without adjoints and even $b = 2a$ the relevant Sasaki-Einstein spaces are the $L^{aaa}$ manifolds. For each of these seed solutions there is a $U(1) \times U(1)$ isometry dual to the flavor symmetry in the dual SCFT on which we can apply the TsT transformation.

## 3.2 TsT

After defining the seed solutions of interest, we are ready to construct a one-parameter family of novel massive type IIA supergravity solutions obtained by applying the TsT transformation in [11].

For both $\mathcal{B} = \mathbf{CP}^2$ and $\mathcal{B} = S^2 \times S^2$ we choose coordinates in which the $U(1) \times U(1)$ symmetry used in the TsT transformation is manifested in terms of the Killing vectors $\partial_{\alpha_2}$ and $\partial_{\alpha_3}$. The TsT transformation consists of a T-duality transformation along $\alpha_2$ followed by a

shift $\alpha_3 \rightarrow \alpha_3 + \gamma \alpha_2$ and then another T-duality transformation along $\alpha_2$. The real parameter $\gamma$ is the supergravity manifestation of an exactly marginal coupling in the dual SCFT. The transformation rules of the NS-NS supergravity sector under this combined action are given explicitly in Appendix C and follow the conventions of [50]. The TsT transformation of the R-R sector is most succinctly written as the following action on the potentials

$$\widetilde{C}_p = C_p + \gamma \left[ C_{p+2} \right]_{[\alpha_2][\alpha_3]} , \tag{48}$$

where the inner product operation $\bullet_{[\alpha_2][\alpha_3]}$ acts on a $p$-form and yields a $p-2$ form[12]

$$(\omega_{p[\alpha_2][\alpha_3]})_{M_1 \dots M_{p-2}} = \omega_{M_1 \dots M_{p-2} \alpha_2 \alpha_3} . \tag{49}$$

The transformed NS-NS sector takes a more complicated form. The TsT transformed metric and B-field for $\mathcal{B} = \mathbf{CP}^2$ are given by

$$
\begin{aligned}
\widetilde{ds}_{10}^2 &= \frac{\sqrt{2 + \cos^2 \alpha}}{3(g^5 m)^{1/3}} \left[ ds_{\text{AdS}_4}^2 + \frac{3 \sin^2 \alpha}{1 + \cos^2 \alpha} ds_{\widetilde{\mathbf{CP}}^2}^2 + \frac{3}{2} d\alpha^2 + \frac{9 \sin^2 \alpha}{2(2 + \cos^2 \alpha)} \mathcal{M} \widetilde{\eta} \widetilde{\eta}^* \right], \\
\widetilde{B}_2 &= \mathcal{M} \Bigg( B_2 + \left( 1 - \mathcal{M}^{-1} \right) \left( \gamma^{-1} d\alpha_2 \wedge d\alpha_3 + \frac{\sin^3 \alpha}{(g^5 m)^{1/3}(1 + \cos^2 \alpha)^2} d\alpha \wedge d\psi \right) \\
&\quad - \frac{3\gamma \mathcal{M}^{-1/2}}{16(g^5 m)^{2/3}} \frac{\sin^4 \alpha \sin^4 \alpha_0 \sin \alpha_1}{1 + \cos^2 \alpha} \left( \sin \alpha_3 \Sigma_1 - \cos \alpha_3 \Sigma_2 \right) \wedge d\psi \Bigg), \\
e^{2\widetilde{\Phi}} &= \mathcal{M} e^{2\Phi},
\end{aligned}
\tag{50}
$$

where a star denotes complex conjugation. The function $\mathcal{M}$ is given by

$$\mathcal{M}^{-1} = 1 + \gamma^2 \frac{\sin^4 \alpha (7 + 5 \cos^2 \alpha + 2 \cos 2\alpha_0 \sin^2 \alpha) \alpha \sin^4 \alpha_0 \sin^2 \alpha_1}{64(g^5 m)^{2/3}(1 + \cos^2 \alpha)^2} , \tag{51}$$

and the modified $\mathbf{CP}^2$ metric is

$$ds_{\widetilde{\mathbf{CP}}^2}^2 = d\alpha_0^2 + \frac{\sin^2 \alpha_0}{4}(\Sigma_1^2 + \Sigma_2^2 + \cos^2 \alpha_0 \Sigma_3^2), \tag{52}$$

where we have defined the modified one-forms

$$
\begin{aligned}
\Sigma_1 &= \cos \alpha_3 d\alpha_1 + \mathcal{M}^{1/2} \sin \alpha_3 \left( \sin \alpha_1 d\alpha_2 - \frac{\gamma}{2g^{5/3} m^{1/3}} \frac{\sin^3 \alpha \sin^2 \alpha_0 \sin \alpha_1}{(1 + \cos^2 \alpha)^2} d\alpha \right), \\
\Sigma_2 &= \sin \alpha_3 d\alpha_1 - \mathcal{M}^{1/2} \cos \alpha_3 \left( \sin \alpha_1 d\alpha_2 - \frac{\gamma}{2g^{5/3} m^{1/3}} \frac{\sin^3 \alpha \sin^2 \alpha_0 \sin \alpha_1}{(1 + \cos^2 \alpha)^2} d\alpha \right), \\
\Sigma_3 &= \mathcal{M}^{1/2} \sigma_3 .
\end{aligned}
\tag{53}
$$

For convenience we have defined the following complex one-form

$$\widetilde{\eta} = \eta + i\gamma \frac{\sqrt{2 + \cos^2 \alpha} \sin^2 \alpha}{4(g^5 m)^{1/3}(1 + \cos^2 \alpha)} \cos \alpha_0 \sin^2 \alpha_0 \sin \alpha_1 \, d\psi . \tag{54}$$

We emphasize that the metric and all fluxes are manifestly real.

---

[12]With a slight abuse of notation, we identify the coordinates $\alpha_{2,3}$ with their index values, i.e. $x^{\alpha_{2,3}} = \alpha_{2,3}$.

When the base space is $\mathcal{B} = S^2 \times S^2$ the NS-NS sector of the TsT transformed solution takes a very similar form.

$$
\begin{aligned}
\widetilde{ds}^2_{10} =& \frac{\sqrt{2+\cos^2\alpha}}{3(g^5m)^{1/3}}\left[ ds^2_{\text{AdS}_4} + \frac{3\sin^2\alpha}{1+\cos^2\alpha} ds^2_{\widetilde{S^2 \times S^2}} + \frac{3}{2}d\alpha^2 + \frac{9\sin^2\alpha}{2(2+\cos^2\alpha)}\mathcal{M}\tilde{\eta}\tilde{\eta}^* \right], \\
\widetilde{B}_2 =& \mathcal{M}\left( B_2 + \left(1-\mathcal{M}^{-1}\right)\left(\gamma^{-1}d\alpha_2 \wedge d\alpha_3 + \frac{\sin^3\alpha}{(g^5m)^{1/3}(1+\cos^2\alpha)^2}d\alpha \wedge d\psi\right) \right. \\
& \left. - \gamma\frac{\sin^4\alpha}{12(g^5m)^{2/3}(1+\cos^2\alpha)}(\cos\alpha_1 \Sigma_1 + \cos\alpha_0 \Sigma_2)\wedge d\psi \right), \\
e^{2\tilde{\Phi}} =& \mathcal{M}e^{2\Phi},
\end{aligned}
\tag{55}
$$

where we have defined

$$
\mathcal{M}^{-1} = 1 + \gamma^2 \frac{\sin^4\alpha((1+\cos^2\alpha)(\cos^2\alpha_1 \sin^2\alpha_0 + \cos^2\alpha_0 \sin^2\alpha_1) + (2+\cos^2\alpha)\sin^2\alpha_0 \sin^2\alpha_1)}{36(g^5m)^{2/3}(1+\cos^2\alpha)^2}, \tag{56}
$$

and the modified $S^2 \times S^2$ metric is given by

$$
ds^2_{\widetilde{S^2 \times S^2}} = \frac{1}{6}\left(d\alpha_0^2 + d\alpha_1^2 + \mathcal{M}(\Sigma_1^2 + \Sigma_2^2)\right), \tag{57}
$$

in terms of the modified one-forms

$$
\begin{aligned}
\Sigma_1 &= \sin\alpha_0\left(d\alpha_2 - \frac{\gamma\cos\alpha_1}{(g^5m)^{1/3}}\frac{\sin^3\alpha}{3(1+\cos^2\alpha)}d\alpha\right), \\
\Sigma_2 &= \sin\alpha_1\left(d\alpha_3 + \frac{\gamma\cos\alpha_0}{(g^5m)^{1/3}}\frac{\sin^3\alpha}{3(1+\cos^2\alpha)}d\alpha\right).
\end{aligned}
\tag{58}
$$

We have again made use of a complex one-form give by

$$
\tilde{\eta} = \eta + i\gamma\frac{\sqrt{2+\cos^2\alpha}\sin^2\alpha}{6(g^5m)^{1/3}(1+\cos^2\alpha)}\sin\alpha_0 \sin\alpha_1 d\psi. \tag{59}
$$

Similarly, for other Kähler-Einstein bases preserving at least $U(1)^2$ global symmetry, the same rules can be applied to find a solution to the equations of motion. Indeed, we checked that this works for the seed solutions associated with the $L^{0b0}$ and $L^{aaa}$ Sasaki-Einstein manifolds dual to the necklace quivers discussed in Section 2. Due to the reduced isometry of the internal space the deformed supergravity solution becomes more complicated and not particularly enlightening so we refrain from presenting it explicitly here.

### 3.3 Quantization and free energy

To ensure that the supergravity solutions discussed above can be promoted to proper string theory backgrounds we should impose quantization of all fluxes threading non-trivial cycles. All the spaces considered above contain a non-trivial six-cycle $M_6$ hence we should impose a flux quantization condition on $F_0$ and $F_6$. As explained in [51], there are various different notions of charge – brane charges, Maxwell charges, Page charges – but only the Page charge is conserved, localized and quantized. However, it is not invariant under large gauge transformations. Therefore, when computing the Page charges, it is important that we consider an appropriate gauge choice for the potentials. Using the potentials for the seed solutions as defined in Appendix B we find the following quantization conditions:

$$
\begin{aligned}
F_0 =& m = \frac{n}{2\pi l_s}, \\
N =& \frac{1}{(2\pi l_s)^5}\int_{M_6}(F_6 + H_3 \wedge C_3) = \frac{1}{(2\pi l_s)^5}\int_{M_6}dC_5 = \frac{16}{3(2\pi \ell_s g)^5}\mathbf{V}_{Y^5},
\end{aligned}
\tag{60}
$$

where $n, N \in \mathbf{Z}$ and $\mathbf{V}_{Y^5}$ is the volume of the five-dimensional Sasaki-Einstein manifold $Y^5$. In the dual field theory, the integer $n$ has the natural interpretation as the sum of all Chern-Simons levels of the gauge groups in the quiver. The integer $N$, on the other hand, is mapped to the rank of the gauge groups.

Next, we want to compute the gravitational $S^3$ free energy which is inversely proportional to the effective four-dimensional Newton's constant $\mathcal{F}_{\text{sugra}} = \frac{\pi L^2}{2G_N^{(4)}}$ [52]. In Einstein frame, the ten-dimensional metric takes the form

$$\mathrm{d}s_{\text{Einstein}}^2 = L^2 e^{2\lambda}(\mathrm{d}s_4^2 + \mathrm{d}s_{M_6}^2), \tag{61}$$

where

$$L^2 = \frac{m^{1/12}}{3 \cdot 2^{5/8} g^{25/12}}, \qquad e^{2\lambda} = \sqrt{\cos(2\alpha) + 3}(\cos 2\alpha + 5)^{1/8}. \tag{62}$$

Consequently, we find the four-dimensional Newton's constant to be

$$\frac{L^2}{G_N^{(4)}} = \frac{L^8}{G_N^{(10)}} \int_{M_6} e^{8\lambda} \mathrm{vol}_6 = \frac{2^{5/2} 3^{5/2} L^8}{5\pi^6 \ell_s^8} \mathbf{V}_{Y^5}, \tag{63}$$

where $16\pi G_N^{(10)} = (2\pi)^7 l_s^8$, the volume form of the internal space with metric $\mathrm{d}s_{M_6}^2$ is given by $\mathrm{vol}_6$ and $\mathbf{V}_{Y^5}$ is the volume of $Y^5$. Using (60) and (63), we thus find that the free energy of our seed solutions is given by

$$\mathcal{F}_{\text{sugra}} = \frac{\pi L^2}{2G_N^{(4)}} = \frac{2^{1/3} 3^{1/6} \pi^3}{5 \mathbf{V}_{Y^5}^{2/3}} n^{1/3} N^{5/3}. \tag{64}$$

The parameters $n$ and $N$ are integer quantized and therefore cannot vary smoothly as a function of the real parameter $\gamma$ that controls the family of TsT deformed solutions we constructed. Indeed, we can explicitly compute these parameters for the family of AdS$_4$ solutions discussed above and find

$$F_0 = m = \frac{n}{2\pi l_s}, \qquad N = \frac{1}{(2\pi l_s)^5} \int_{S^6} d\widetilde{C}_5. \tag{65}$$

Since $\widetilde{C}_9 = C_9$, the Romans mass $F_0$, $n$ and therefore $k$ remain unchanged. The potential $C_5$ on the other hand does change under the TsT transformation, but it does so by a globally well-defined differential form. Therefore, its integral cohomology class does not change and $N$ remains identical after the transformation. Since the TsT deformed solutions are still of the form (61) we can proceed analogously to the undeformed case to compute the free energy. As is clear from (50) and (55), in string frame, the warp factor in front of AdS$_4$ is unaltered by the TsT transformation, while the dilaton picks up a factor of $\mathcal{M}$. The relation between string and Einstein frame, $ds_{\text{string}}^2 = e^{\Phi/2} ds_{\text{Einstein}}^2$, then implies the transformation rule $L^8 e^{8\lambda} \to \mathcal{M}^{-1} L^8 e^{8\lambda}$. The volume form of the internal manifold however transforms as $\mathrm{vol}_6 \to \mathcal{M}\mathrm{vol}_6$ and therefore it follows that the $S^3$ free energy is unaffected by the TsT transformation. This is in line with the expectation that $\gamma$ corresponds to an exactly marginal deformation in the field theory which leaves the $S^3$ free energy invariant. This constitutes a non-trivial consistency check of our supergravity analysis.

We conclude our discussion with some explicit calculations of the $S^3$ free energy using the holographic result above for the solutions dual to the quiver gauge theories discussed in Section 2. As emphasized in [16], the calculation is streamlined by the expression in (64) which makes it clear that the holographic $S^3$ free energy depends only on the volume of the associated Sasaki-Einstein manifold $Y^5$. Using the standard expression for the volume of $S^5$,

as well as the volumes of $T^{1,1}$ and $L^{aba}$ computed in [53, 54], we have compiled the results for the holographic free energy in Table 1. Comparing these results with the large $N$ $S^3$ free energy of the various quiver theories presented in Section 2 we indeed find agreement for all cases by identifying $n = bk$ where $b$ is the number of gauge nodes in the quiver. The only case not covered by this analysis is the supergravity solution dual to the one node theory with $\mathcal{N} = 3$ supersymmetry. This example was studied holographically in some detail in [26] where it was shown that indeed the supersymmetric localization and holographic results for the $S^3$ free energy agree.

Finally we point out that the holographic manifestation of the RG flows discussed around (33), for $b = 2a \geq 2$, should be in terms of supergravity solutions which interpolate between the AdS$_4$ vacua associated with $L^{02a0}$ in the UV and $L^{aaa}$ in the IR. Indeed from the results in Table 1 one finds

$$\frac{\mathcal{F}_{L^{aaa}}}{\mathcal{F}_{L^{02a0}}} = \left(\frac{27}{32}\right)^{2/3}. \tag{66}$$

This confirms the expectation from field theory and gives further evidence that indeed such RG flows exists.

Table 1: The holographic $S^3$ free energy for various choices of internal manifolds.

| $\mathcal{B}$ | $Y^5$ | $\mathbf{V}_{Y^5}$ | $\mathcal{F}_{\text{sugra}}$ |
|---|---|---|---|
| $\mathbf{CP}^2$ | $S^5$ | $\pi^3$ | $\frac{2^{1/3}3^{1/6}}{5}\pi n^{1/3}N^{5/3}$ |
| $S^2 \times S^2$ | $T^{1,1}$ | $\frac{16\pi^3}{27}$ | $\frac{3^{13/6}}{5\cdot2^{7/3}}\pi n^{1/3}N^{5/3}$ |
| | $L^{0b0}$ | $\frac{\pi^3}{b}$ | $\frac{2^{1/3}3^{1/6}b^{2/3}}{5}\pi n^{1/3}N^{5/3}$ |
| | $L^{\frac{b}{2}\frac{b}{2}\frac{b}{2}}$ | $\frac{32\pi^3}{27b}$ | $\frac{3^{13/6}b^{2/3}}{5\cdot2^3}\pi n^{1/3}N^{5/3}$ |

# 4 Discussion

In this note we studied some examples of 3d $\mathcal{N} = 2$ Chern-Simons-matter theories associated to 4d $\mathcal{N} = 1$ parent quiver gauge theories. These theories are constructed by interpreting the 4d quiver representation and superpotential in a 3d context and assigning equal Chern-Simons levels $k$ to all gauge nodes. Our focus was on exploring the space of exactly marginal deformations of these theories and their dual holographic manifestation. In particular, we constructed novel one-parameter families of massive type IIA AdS$_4$ supergravity backgrounds dual to a one-dimensional submanifold of the conformal manifold of these 3d $\mathcal{N} = 2$ SCFTs.

Notably, the one-parameter families of supergravity solutions we found depend on one real parameter whereas the marginal couplings in the 3d SCFTs that preserve U(1) × U(1) flavor symmetry are complex. Using only the TsT transformation in supergravity we cannot generate such a complex parameter. One can try to introduce a complex parameter in the supergravity construction by considering a Ts$_\gamma$S$_\sigma$T-transformation where the S$_\sigma$ denotes an S-duality transformation in the intermediate type IIB solution. When performing an S-duality transformation, a priori, $\sigma \in$ SL(2, **R**) has three parameters. Defining the complex axion-dilaton as $\tau = C_0 + ie^{-\phi}$, a general S-duality acts on $\tau$ and the $B_2, C_2$ doublet as follows,

$$\tau \to \frac{\mathfrak{a}\tau + \mathfrak{b}}{\mathfrak{c}\tau + \mathfrak{d}}, \qquad \text{with} \quad \mathfrak{a}\mathfrak{d} - \mathfrak{b}\mathfrak{c} = 1, \tag{67}$$

and

$$\begin{pmatrix} C_2 \\ B_2 \end{pmatrix} \to \begin{pmatrix} \mathfrak{a} & \mathfrak{b} \\ \mathfrak{c} & \mathfrak{d} \end{pmatrix} \begin{pmatrix} C_2 \\ B_2 \end{pmatrix}. \tag{68}$$

This transformation will generate a larger family of $AdS_4$ solutions in IIA supergravity when applied to a given seed solution. However to ensure that these backgrounds are dual to the complex marginal deformations of interest the quantized fluxes $n$ and $N$ need to be the same as the ones for the original seed solution. We have shown above that this condition is obeyed by the TsT transformation. However, for the more general $Ts_\gamma S_\sigma T$ transformation not all $\sigma \in SL(2, \mathbf{R})$ are compatible with this condition. In order for the Romans mass in (60) to remain the same $\sigma$ has to be of the form

$$\sigma = \begin{pmatrix} \pm 1 & \mathfrak{b} \\ 0 & \pm 1 \end{pmatrix}, \tag{69}$$

i.e. only S-duality transformations of the form $\tau \to \tau + \mathfrak{b}$, $\mathfrak{b} \in \mathbf{R}$ are allowed. This leaves us with a transformation that depends on the 2 real parameters $\gamma$ and $\mathfrak{b}$. Under this combined $Ts_\gamma S_\mathfrak{b} T$ transformation the RR potentials of the supergravity solution are modified with respect to the pure TsT transformation but all gauge invariant quantities remain unchanged. It is therefore unclear to us whether the parameter $\mathfrak{b}$ should really be viewed as generating a new distinct supergravity solution. As discussed in Section 2.3 a similar puzzle exists for the ABJM SCFTs and the TsT transformation of the $AdS_4 \times S^7$ solution of 11d supergravity, namely the supergravity family of solutions with $U(1)^4$ isometry found in [11] depend on a real parameter whereas the exactly marginal coupling of the ABJM theory compatible with this symmetry is complex. It will be most interesting to find a resolution to this apparent puzzle.

Our work should be viewed only as a starting point in the exploration of the conformal manifolds of this class of 3d $\mathcal{N} = 2$ quiver gauge theories and their dual supergravity description. There are a number of interesting open questions and possible generalizations and we list some of them below

- The $AdS_4$ supergravity solutions we constructed in Section 3 should fall into the general classification of supersymmetric $AdS_4$ vacua of massive IIA supergravity presented in [55]. It will be very interesting to flesh this out in more detail and perhaps extract important lessons that will allow for the construction of more general classes of solutions. It will also be interesting to understand how our supergravity construction fits into the broader effort to understand the exceptional/generalized geometry that underlies the properties of supergravity solutions dual to SCFT marginal deformations, see [12, 56]. In particular, it is clear from the SCFT discussions in Section 2 that there should be more general families of supergravity solutions dual to the marginal deformations that break the U(1)×U(1) flavor symmetry preserved by the TsT transformation. Constructing these solutions will be challenging by direct methods and perhaps the techniques discussed in [12, 56] may shed some light on this problem. We note that it should be possible to calculate the expectation values of supersymmetric Wilson lines in the 3d SCFTs using the supergravity solutions we construct. In [16] this quantity was computed at large $N$ for the class of seed supergravity solutions we presented above. Supersymmetric localization suggests that this Wilson line vev should not depend on the exactly marginal coupling and it will be interesting to confirm this by an explicit probe string calculation in the family of new $AdS_4$ solutions constructed in Section 3. Finally, it is desirable to develop tools to compute the KK spectrum of the supergravity solutions we constructed since this will elucidate how the spectrum of operator dimensions in the dual SCFT depends on the exactly marginal coupling corresponding to the deformation parameter $\gamma$. The recent results in [22] may serve as a starting point for such analysis.

- While we have computed the dimensions of the conformal manifolds for the class of theories we discussed in Section 2 we have not discussed most of the interesting questions pertaining to their physics. It will be very interesting to understand what is the global structure of these conformal manifolds. Presumably these conformal manifolds are intrinsically strongly coupled and are therefore compact, see [5] for a general recent discussion. It is important to elucidate the connection between the conformal manifolds of the 3d $\mathcal{N} = 2$ SCFTs and that of the parent 4d $\mathcal{N} = 1$ quiver gauge theory. It is also important to develop tools to calculate physical observables, like the spectrum of operators, OPE coefficients, and the Zamolodchikov metric, along these conformal manifolds. It will also be nice to calculate explicitly the recently discussed nilpotency index of the conformal manifold, see [57]. The one node quiver with three chiral adjoints discussed in Section 2.1 has a superpotential similar to that of the 3d $\mathcal{N} = 2$ XYZ model and its exactly marginal deformation. This conformal manifold has been studied in some detail in [58] and it will be interesting to understand if there is any relation between the physics of these two models.

- While our main focus here was on exactly marginal deformations, in Section 2.2 we also briefly discussed the existence of what appears to be universal RG flows in this class of 3d $\mathcal{N} = 2$ SCFTs. It is important to understand the physics of these RG flows and how the universal relation between the UV and IR $S^3$ free energy is modified for finite $N$. There should also be supergravity domain wall solutions that interpolate between the AdS$_4$ vacua dual to the UV and IR SCFTs and it will be nice to construct them explicitly.

- Finally, we note that there are more general classes of 4d $\mathcal{N} = 1$ quiver gauge theories which can be used as "parents" for 3d $\mathcal{N} = 2$ SCFTs by following the same pattern as in Section 2. The conformal manifolds for some of these 4d $\mathcal{N} = 1$ SCFTs was studied in [28] and it should be possible to generalize this analysis to 3d both in the SCFT and in the holographically dual AdS$_4$ solutions.

We hope that our work will stimulate further explorations of these interesting questions.

## Acknowledgements

We are grateful to the anonymous referee for the careful reading of the manuscript which led to clarifying several aspects of the discussion in Section 2.

**Funding information**    The work of NB is supported in part by the Odysseus grant G0F9516N from the FWO. PB is supported by the STARS-StG grant THEsPIAN. FFG was a Postdoctoral Fellow of the Research Foundation - Flanders (FWO) during the initial stages of this work and is currently supported by the University of Iceland Recruitment Fund. The work of VSM was supported by a doctoral fellowship from the FWO. This work is also supported in part by the KU Leuven C1 grant ZKD1118 C16/16/005.

# A  Supergravity conventions

The solutions constructed in this note solve the equations of motion of ten-dimensional massive type IIA supergravity [59]. The bosonic field content consists of a metric $G_{MN}$, a dilaton $\Phi$, a the three-form $H_3$ and $n$-form field strengths $F_n$, with $n = 0, 2, 4$. The Romans mass, $F_0$, does not have any propagating degrees of freedom. The fermionic field content consists of a doublet of gravitinos, $\psi_M$, and a doublet of dilatinos, $\lambda$. The components of these doublets are of opposite chirality.

In this note we use the "democratic formalism" in which the number of R-R fields is doubled such that $n$ runs over $0, 2, 4, 6, 8, 10$ [60]. This redundancy is removed by introducing duality conditions for all R-R fields

$$F_n = (-1)^{\frac{(n-1)(n-2)}{2}} \star_{10} F_{10-n}. \tag{70}$$

These duality conditions should be imposed by hand after deriving the equations of motion from the action. The bosonic part of the action written in string frame is given by

$$S_{\text{bos}} = \frac{1}{2\kappa_{10}^2} \int \star_{10} \Big[ e^{-2\Phi} \Big( R + 4|d\Phi|^2 - \frac{1}{2}|H_3|^2 \Big) - \frac{1}{4} \sum_n |F_n|^2 \Big], \tag{71}$$

where the ten-dimensional Newton constant $\kappa_{10}$ is related to the string length through $4\pi\kappa_{10} = (2\pi l_s)^8$ and we have defined

$$\star_{10} |A|^2 \equiv \star_{10} \frac{1}{n!} A_{M_1 \dots M_n} A^{M_1 \dots M_n} = \star_{10} A \wedge A. \tag{72}$$

This action should be completed by its fermionic counterpart, which we do not write explicitly. The Bianchi identities and equations of motion derived from the action (71) are

$$dH_3 = 0, \qquad \text{and} \qquad d(e^{-2\Phi} \star_{10} H_3) + \frac{1}{2} \sum_n \star_{10} F_n \wedge F_{n-2} = 0, \tag{73}$$

for the NS-NS field $H_3$ and

$$dF_n - H_3 \wedge F_{n-2} = 0, \tag{74}$$

for the R-R form fields. The dilaton and the Einstein equations of motion can be written as

$$\begin{aligned}
0 &= \nabla^2 \Phi - |d\Phi|^2 + \frac{1}{4}R - \frac{1}{8}|H_3|^2, \\
0 &= R_{MN} + 2\nabla_M \nabla_N \Phi - \frac{1}{2}|H_3|_{MN}^2 - \frac{1}{4}e^{2\Phi} \sum_n |F_n|_{MN}^2,
\end{aligned} \tag{75}$$

where we have defined

$$|A_n|_{MN}^2 \equiv \frac{1}{(n-1)!} (A_n)_M{}^{M_2 \dots M_n} (A_n)_{NM_2 \dots M_n}. \tag{76}$$

# B  Potentials in the democratic formalism

The type IIA fluxes we use are defined as follows

$$F_p = dC_{p-1} - H_3 \wedge C_{p-3} + F_0 \, e^{B_2}\big|_p, \tag{77}$$

where $e^{B_2}\big|_p$ denotes the $p$-form term in the expansion of the exponential. In these conventions the potentials for the field strengths in equation (43) are given explicitly by

$$B_2 = \left(\frac{1}{g^5 m}\right)^{1/3} \mathrm{d}\left(\frac{\cos\alpha}{1+\cos^2\alpha}\right) \wedge \eta\,,$$

$$C_1 = -\left(\frac{m^2}{g^5}\right)^{1/3} \frac{\cos\alpha\sin^2\alpha}{2(1+\cos^2\alpha)(2+\cos^2\alpha)}\eta\,,$$

$$C_3 = \left(\frac{m}{g^{10}}\right)^{1/3} \left(\frac{\omega_3}{\sqrt{3}} + \frac{(2+3\cos^2\alpha)\sin^4\alpha}{4(1+\cos^2\alpha)^2}\eta\wedge J\right)\,,$$

$$C_5 = \frac{\cos\alpha}{g^5}\left(\frac{2+\cos^2\alpha}{6\sqrt{3}}\eta\wedge\mathrm{vol}_{\mathrm{AdS}_4} - \frac{2}{\sqrt{3}(1+\cos^2\alpha)}\omega_3\wedge J\right.$$
$$\left. - \frac{39+59\cos^2\alpha+21\cos^4\alpha+9\cos^6\alpha}{12(1+\cos^2\alpha)^3}J\wedge J\wedge\eta\right)\,,$$

$$C_7 = \left(\frac{1}{g^{20}m}\right)^{\frac{1}{3}}\left(\frac{2\cos^2\alpha}{\sqrt{3}(1+\cos^2\alpha)^2}\omega_3\wedge J\wedge J - \frac{2+5\cos^2\alpha-\cos^4\alpha}{12\sqrt{3}(1+\cos^2\alpha)}\mathrm{vol}_{\mathrm{AdS}_4}\wedge\eta\wedge J\right)\,,$$

$$C_9 = \left(\frac{1}{g^{25}m^2}\right)^{1/3}\cos\alpha\frac{40+50\cos^2\alpha-9\cos^4\alpha+2\cos^6\alpha+\cos^8\alpha}{60\sqrt{3}(1+\cos^2\alpha)^2}\mathrm{vol}_{\mathrm{AdS}_4}\wedge J\wedge J\wedge\eta\,.$$

(78)

Here we introduced the three-form $\omega_3$ whose exterior derivative gives the volume form on $\mathrm{AdS}_4$, $\mathrm{d}\omega_3 = \mathrm{vol}_{\mathrm{AdS}_4}$.

## C  TsT transformation

In this appendix we present an algorithmic procedure to obtain the TsT transformation of a type IIA(B) supergravity background with metric $G_{MN}$, dilaton $\Phi$, anti-symmetric tensor field $B_{MN}$ with three-form field strength $H_3$, and modified R-R field strengths $F_p = \mathrm{d}C_{p-1} - \mathrm{d}B \wedge C_{p-3}$ with $p$ odd for type IIB and $F_p = \mathrm{d}C_{p-1} - \mathrm{d}B \wedge C_{p-3} + m\,e^B$ with $p$ even for type IIA. This appendix follows closely the exposition in [50].

Let us start by defining the composite object $e_{MN} = G_{MN} + B_{MN}$. Next, assume that the coordinates $\alpha_{2,3}$ parameterize the two commuting U(1) isometries of the solution. The TsT transformation along $\alpha_2$ and $\alpha_3$ now consists of three steps. First, a T-duality along $\alpha_2$. The type IIA(B) solution now becomes a type IIB(A) solution with dual coordinates $\{\tilde\alpha_2, \tilde\alpha_3\}$. Next, shift the $\tilde\alpha_3$ coordinate in the T-dual solution as

$$\tilde\alpha_3 \to \tilde\alpha_3 + \gamma\tilde\alpha_2\,, \tag{79}$$

with $\gamma$ an arbitrary real parameter. Finally, perform another T-duality along $\tilde\alpha_2$, going back to type IIA(B). For simplicity, we call the final coordinates again $\{\alpha_2, \alpha_3\}$. Starting from a solution to the equations of motion, this transformation is guaranteed to result in a new supergravity solution. Furthermore, it will not generate new singularities.

Carefully applying the Buscher rules [61,62], the NS-NS fields $\widetilde{G}_{MN}, \widetilde{B}_{MN}$ and $\tilde\Phi$ of the TsT transformed solution can be obtained from the NS-NS fields $G_{MN}, B_{MN}$ and $\Phi$ of the original solution from the following rule:

$$\tilde{e}_{MN} = \mathcal{M}\left\{e_{MN} - \gamma\left[\det\begin{pmatrix} e_{\alpha_2\alpha_3} & e_{\alpha_2 N} \\ e_{M\alpha_3} & e_{MN} \end{pmatrix} - \det\begin{pmatrix} e_{\alpha_3\alpha_2} & e_{\alpha_3 N} \\ e_{M\alpha_2} & e_{MN} \end{pmatrix}\right] + \gamma^2\det\begin{pmatrix} e_{\alpha_2\alpha_2} & e_{\alpha_2\alpha_3} & e_{\alpha_2 N} \\ e_{\alpha_3\alpha_2} & e_{\alpha_3\alpha_3} & e_{\alpha_3 N} \\ e_{M\alpha_2} & e_{M\alpha_3} & e_{MN} \end{pmatrix}\right\}\,,$$

$$e^{2\tilde\Phi} = \mathcal{M}e^{2\Phi}\,,$$

(80)

where $\tilde{e}_{MN} = \widetilde{G}_{MN} + \widetilde{B}_{MN}$ and $\mathcal{M}$ is defined as

$$\mathcal{M} = \left\{ 1 - \gamma(e_{\alpha_2\alpha_3} - e_{\alpha_3\alpha_2}) + \gamma^2 \det\begin{pmatrix} e_{\alpha_2\alpha_2} & e_{\alpha_2\alpha_3} \\ e_{\alpha_3\alpha_2} & e_{\alpha_3\alpha_3} \end{pmatrix} \right\}^{-1}. \tag{81}$$

The transformed R-R potentials $\widetilde{C}_p$ can be computed from the original ones as

$$\widetilde{C}_p = C_p + \gamma [C_{p+2}]_{[\alpha_2][\alpha_3]}, \tag{82}$$

where the inner product operation $\bullet_{[\alpha_2][\alpha_3]}$ acts on a $p$-form and returns a $p-2$ form

$$(\omega_{p_{[\alpha_2][\alpha_3]}})_{M_1...M_{p-2}} = \omega_{M_1...M_{p-2}\alpha_2\alpha_3}. \tag{83}$$

One can check explicitly that this transformation is indeed a solution generating procedure such that it transforms one set of fields solving the equations of motion into a new one.

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
