# Peer review of "Marginal deformations from type IIA supergravity"

_SciPost Physics, doi:SciPost Phys. 10, 140 (2021)_

## Round 2 · Referee Report · Anonymous (Referee 1) · 2021-4-22

Strengths

1 - Construction of novel AdS4 solutions dual to marginal deformations of known 3d SCFTs
2 - Successful holographic match of field theory and supergravity results
3 - Section 3 (Supergravity) is very clearly explained
4 - Starting point for several potential generalizations, clearly laid out in the conclusions.

Weaknesses

1 - Several confusing or unclear statements and calculations in section 2 (Field Theory)

Report

This note combines dual field theory and supergravity analysis to study marginal deformations of a class of three-dimensional N=2 SCFTs first studied by Guarino, Jafferis and Varela, as well as IR fixed points of relevant deformations of those theories. Conformal manifolds of 3d N=2 SCFTs are generically expected to lack weak coupling cusps and hence be compact. The study of global aspects of the conformal manifold in field theory is challenging, therefore the holographic approach advocated in this paper can provide valuable new information.

The paper consists mainly of two sections. In section 2 (Field theory) the relevant field theories are introduced, their symmetries are discussed and the dimension of their conformal manifolds is calculated, following standard methods. In section 3 (Supergravity) a family of novel AdS4 solutions dual to a one (real) parameter family of marginal deformations of the SCFTs are constructed using the TsT transformation of Lunin and Maldacena. The large-N three-sphere free energy is calculated holographically, matching the large-N limit of the field theory result which is obtained by supersymmetric localisation. The note is concluded by section 4 (Discussion), which lays out very clearly some open questions and several future research directions.

While not groundbreaking, the paper contains a number of interesting results and paves the way for a more detailed study of marginal deformations of 3d N=2 SCFTs using holography. This makes the paper potentially worthy of publication. Before recommending publication, I would ask the authors to go through a minor revision to address some weaknesses in the field theory section.

Requested changes

1 - Page 7: explain what is the SU(2) flavor symmetry not manifest at the level of the superpotential. 2 - Page 7: clarify whether the counting in (12) is done in the theory with vanishing superpotential or with superpotential (10). If the latter, discuss chiral ring relations. (This applies to several points below.) 3 - Page 7: revisit the discussion of abelian flavor symmetries to avoid overcounting. Make sure that no linear combination is a gauge symmetry or the Cartan of SU(2)_F. 4 - Page 8: isn't the baryonic symmetry gauged? Discuss the chiral ring relation for the dimension count. 5 - Page 9: please revisit the 3 U(1) flavor symmetries below (23). Again a linear combination appears to be a gauge symmetry. 6 - Page 9 (bottom): clarify whether the large flavor symmetry is in the presence or absence of the superpotential (25). Please also double check the linear combinations in the last line. 7 - Page 10 (top): aren't the baryonic symmetries gauged? 8 - Page 12: most of the discussion in section 2.3.2 (and 2.3.3) seems to ignore the dependence of the electric charge of the monopole operators on the Chern-Simons level k. Please amend the discussion so that it applies to both k=1 and k=2. 9 - Page 13: correct typo `ration' after (42). 10 - Page 15: please explain what is sigma in equations (45), (47). Is it the contact one-form previously denoted by eta? 11 - Page 22: do the indices 1 and 2 in equation (80) denote the coordinates \alpha_2 and \alpha_3? Please clarify.

  • validity: high
  • significance: good
  • originality: good
  • clarity: good
  • formatting: excellent
  • grammar: excellent

Author:  Pieter Bomans  on 2021-04-29  [id 1389]

(in reply to Report 1 on 2021-04-22)

We are grateful to the referee for the careful and detailed comments. We agree that some parts of the discussion in Section 2 are not well explained and can lead to confusion or wrong interpretation. Below we briefly address the 11 comments and questions raised by the referee.

  • Comments 1-3 refer to the discussion on page 7. We performed the chiral operator counting in Equation (11) and analyzed the corresponding global symmetries for the theory with no superpotential. We will clarify this point in an updated version of the manuscript. In addition, we plan to clarify the discussion to emphasize the distinction between the theory with ${\rm U}(N)\times {\rm U}(N)$ gauge group and a topological ${\rm U}(1)$ global symmetry and the theory with ${\rm SU}(N)\times {\rm SU}(N)$ gauge group and a baryonic ${\rm U}(1)$ global symmetry. We believe that these clarifications will address comments 1, 2, and 3 by the referee.
  • Comment 4. The baryonic symmetry is not gauged for the theory with ${\rm SU}(N)\times{\rm SU}(N)$ gauge group and, as the referee states, is indeed gauged for the ${\rm U}(N)\times{\rm U}(N)$ theory. In both cases however there is a ${\rm U}(1)$ global symmetry (baryonic for the ${\rm SU}(N)\times{\rm SU}(N)$ theory and topological for the ${\rm U}(N)\times{\rm U}(N)$ theory) and thus the counting of exactly marginal operators in Equation (18) remains the same. When counting the chiral operators in (17) we have taken the chiral ring relations into account. We will clarify this point further in the revised version.
  • Comments 5-7. Here again we analyzed global symmetries and counted chiral operators in the theory with no superpotential. This is similar to the discussion above about Comments 1-3. We will clarify this in some detail in the revised version.
  • Comment 8. The referee correctly points out that some of the discussion in Sections 2.3.2 and 2.3.3 applies only to the ABJM theory with $k=1$ and has to be clarified for the theory with $k=2$. For instance the form of the schematic relations in Equation (35) is slightly modified for $k=2$ and the superpotential in Equation (39) has to change to ${\rm Tr}(A_1 T A_1)$. We will add an appropriate discussion about this in the revised version of the manuscript. We note that the counting of exactly marginal operators in Equations (38) and (41) is not affected by these minor changes.
  • Comment 9. The typo will be corrected.
  • Comment 10. As discussed in the text below Equation (43) $\eta$ and $\sigma$ are not identical and are related as $\eta = {\rm d}\psi+\sigma$. We will add a comment about this around Equations (45) and (47).
  • Comment 11. Yes, this is what we mean by these indices. We will clarify this in the updated version.

Best regards, Nikolay, Pieter and Fridrik

Anonymous on 2021-06-01  [id 1481]

(in reply to Pieter Bomans on 2021-04-29 [id 1389])
Category:
remark

I thank the authors for the detailed reply to my previous report and their changes in the revised version. I no longer have concerns about the QFT part, which is now presented clearly. The paper is interesting and opens up several interesting directions for future work. I recommend publication.

---

## Round 2 · Referee Report · Anonymous (Referee 2) · 2021-4-30

Strengths

This paper is well-written. It addresses a problem of interest. The work is localised to a particular set of models, that are clearly specified. The analysis is good and nicely explained. The checks between supergravity and field theory (for example for the RG flow proposed) are solid. The supergravity backgrounds are clearly written and aspects of the SCFTs are cleanly explained.
I find both the supergravity solution to be very clearly presented and the TsT compactly written in the appendix.

I liked this idea of Ts_{\gamma}S_{\sigma}T!

Weaknesses

There are some minor imprecisions in the CFT section regarding the transformation of some of the fields. These do not majorly affect the analysis. I believe they were pointed out by another referee.

I hoped the authors were finding the RG flow of section 2.2 in supergravity.
May be this is not so easy task. I encourage the authors to address this point in a future publication.

Report

I believe the paper should be published by this journal. As stated, it is a good contribution. Clearly written and with a couple of interesting checks of its claims. The minor imprecisions pointed by the previous referee can be easily addressed. Please do so.

As a suggestion for the authors: would it be possible to study some RG-flows of the 3d SCFTs (not necessarily ending in an IR fixed point), by importing flows in the 4d set-ups. Something along the lines of Klebanov-Tseytlin-Strassler.

Question for the authors: any field theoretical clue about where is the factor
(27/32)^(2/3)
is coming from? In particular the power of 2/3?
This is probably hard to answer in QFT, whilst in gravity is quite transparent.

Requested changes

I do not have major requests, except from addressing those minor imprecise things in the QFT section.

  • validity: high
  • significance: good
  • originality: good
  • clarity: high
  • formatting: excellent
  • grammar: perfect

Author:  Pieter Bomans  on 2021-05-24  [id 1462]

(in reply to Report 2 on 2021-04-30)

We would like to thank Referee 2 for the positive report.

We agree that the questions for future work raised by the referee are interesting and deserve further study.
The supergravity domain walls that capture some of the conformal and non-conformal phases of some of the 3d N=2 theories we discuss are studied using 4d supergravity techniques in 1605.09254, but as the referee points out there is more to explore. However, for the quiver theories with two or more nodes these questions have to be explored directly in 10d supergravity since there is no known 4d consistent truncation.
The power 2/3 that appears in the relation between the large N central charge of the parent 4d N=1 SCFT and the $S^3$ free energy of the 3d N=2 SCFT was determined by supersymmetric localization in 1507.05817. As the referee suggests, this relation needs to be understood better, in particular when it comes to its validity beyond the large N approximation.

Best regards,
Nikolay, Pieter, Vincent and Fridrik

Anonymous on 2021-05-24  [id 1463]

(in reply to Pieter Bomans on 2021-05-24 [id 1462])
Category:
remark

I do agree with the comments of the authors. On my side, this paper is ready for publication. As the authors correctly understood, my comments are mostly for 'future work'.

---

## Editorial Decision

published